

# Supersolid phase of a spin-orbit-coupled Bose-Einstein condensate: A perturbation approach

**Giovanni Italo Martone[1,2⋆] and Sandro Stringari[3]**

**1** Université Paris-Saclay, CNRS, LPTMS, 91405 Orsay, France
**2** Laboratoire Kastler Brossel, Sorbonne Université, CNRS,
ENS-PSL Research University, Collège de France; 4 Place Jussieu, 75005 Paris, France
**3** INO-CNR BEC Center and Dipartimento di Fisica, Università di Trento, I-38123 Povo, Italy

⋆ giovanni_italo.martone@lkb.upmc.fr

## Abstract

The phase diagram of a Bose-Einstein condensate with Raman-induced spin-orbit coupling includes a stripe phase with supersolid features. In this work we develop a perturbation approach to study the ground state and the Bogoliubov modes of this phase, holding for small values of the Raman coupling. We obtain analytical predictions for the most relevant observables (including the periodicity of stripes, sound velocities, compressibility, and magnetic susceptibility) which are in excellent agreement with the exact (non perturbative) numerical results, obtained for significantly large values of the coupling. We further unveil the nature of the two gapless Bogoliubov modes in the long-wavelength limit. We find that the spin branch of the spectrum, corresponding in this limit to the dynamics of the relative phase between the two spin components, describes a translation of the fringes of the equilibrium density profile, thereby providing the crystal Goldstone mode typical of a supersolid configuration. Finally, using sum-rule arguments, we show that the superfluid density can be experimentally accessed by measuring the ratio of the sound velocities parallel and perpendicular to the direction of the spin-orbit coupling.

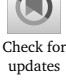

# 1 Introduction

Bose-Einstein condensates (BECs) with synthetic spin-orbit coupling have been the subject of intense investigations in the last decade (see the reviews [1–8] and references therein). They represent an ideal framework for observing new quantum phenomena, resulting from the interplay between the modified single-particle spectrum and the interaction. One of the most intriguing configurations appearing in the phase diagram of a spin-orbit-coupled BEC is the so-called stripe phase, where the atoms condense in a superposition of states with finite momenta. Its importance lies in being a crystalline structure with superfluid properties, meaning that it exhibits the phenomenon of supersolidity (see the review [9]). This feature prompted important efforts towards the experimental observation of the supersolid behavior, which was eventually first reported in a spin-orbit-coupled configuration [10] and in a BEC gas inside optical resonators [11, 12]. A later study on the stripe phase of BECs with Raman-induced spin-orbit coupling showed the presence of long-range coherence between the different spin components [13]. More recently, the supersolid phase has been identified in a series of experiments with dipolar Bose gases, in the form of linear [14–16] and two-dimensional [17, 18] droplet arrays; in such systems the investigation of collective modes [19–22], rotational and superfluid properties [23–25], out-of-equilibrium phase coherence effects [26], and the role of temperature [27] represents a very active field of research. Although the above experiments were carried out in three dimensions, the appearance of a stripe phase was also predicted by Monte Carlo calculations in two-dimensional configurations, where the polarization direction of the dipoles is tilted with respect to the axis perpendicular to the plane in which they are confined; however, it is still a subject of debate whether such stripes are superfluid or

not [28, 29].

It is well theoretically understood that a common feature of the excitation spectra of supersolids is the presence of two or more Goldstone modes [30–42]. In general, one of them is associated with the superfluid motion of particles through the crystalline structure, which stays at rest. The other modes involve oscillations of the density fringes about the equilibrium position, which are typical of crystalline solid bodies. Actually, the superfluid and crystal characters completely separate only in the limit of infinite wavelength; far from this limit they are both present to some extent, leading to mode hybridization [19–21]. This physics is well understood in single-component systems, such as dipolar gases or BECs in optical resonators. Spin-orbit-coupled BECs in the stripe phase are of special interest because of their additional spin degree of freedom. Its role is well understood in Bose-Bose mixtures without spin-orbit coupling, whose Bogoliubov spectrum is made of a density and a spin branch [43, 44]. In the infinite-wavelength limit the density and spin branches are both gapless and correspond to the dynamics of the total and relative phase between the two spin components, respectively. In the presence of spin-orbit coupling the nature of the spin excitations is instead far from being obvious. On the one hand, one expects that the spin modes are gapped as a consequence of the Raman coupling. On the other hand, a second gapless mode is predicted to occur in the stripe phase [36] as a result of the spontaneous breaking of translational invariance.

The main purpose of this paper is to gain insight into the Goldstone modes of a spin-orbit-coupled BEC in the stripe phase, and in particular to point out the spin nature of the gapless crystal mode. This aspect has been recently numerically investigated in trapped configurations [45]. Here we consider homogeneous setups where analytic results can be obtained in an explicit way. For concreteness we will focus on BECs featuring a combination of the Rashba [46] and Dresselhaus [47] spin-orbit coupling with equal weights. Such systems can be realized by coupling a condensate to the light field generated by a pair of Raman lasers, detuned in frequency and with orthogonal polarizations. This setup was first employed by Ian Spielman's group at NIST [48]. When the Raman coupling term between the two spin components is zero the ground state of the system corresponds to an unpolarized BEC mixture. We will carry out our analysis in the limit where the Raman coupling is small, and study how this perturbation modifies the properties of the ground state and the Bogoliubov modes, particularly those in the phonon regime. Further information about the compressibility, magnetic susceptibility, and superfluid density will be obtained combining the perturbation approach with the method of sum rules.

This work is structured as follows. In Sec. 2 we briefly present the model describing a Raman-induced spin-orbit-coupled BEC and review its main properties, with a special focus on the stripe phase. Section 3 deals with the results of our perturbation approach for a BEC at equilibrium in the stripe phase, providing analytic formulas for several important observables. The perturbation method is subsequently extended to the study of the Bogoliubov modes in Sec. 4: we compute the velocities of sound waves and the corresponding amplitudes, the most relevant sum rules, and the superfluid density. We summarize in Sec. 5. The details of the calculations are reported in the Appendices.

## 2 The model

In this section we briefly introduce the model describing the system under consideration. Section 2.1 is devoted to the single-particle physics, while in Sec. 2.2 we summarize the properties of the many-body ground state obtained including interactions within the mean-field approach. More details about the stripe phase are discussed in Sec. 2.3.

## 2.1 Single-particle physics

A three-dimensional (3D) spin-1/2 BEC with Raman-induced spin-orbit coupling is described by the single-particle Hamiltonian

$$h_{\text{SO}} = \frac{(p_x - \hbar k_R \sigma_z)^2}{2m} + \frac{p_\perp^2}{2m} + \frac{\hbar \Omega_R}{2} \sigma_x \,. \tag{1}$$

Here $m$ is the atom mass, $p_\perp^2 = p_y^2 + p_z^2$, and $\sigma_j$ ($j = x, y, z$) denote the usual $2 \times 2$ Pauli matrices. The Raman recoil momentum $\hbar k_R$ plays the role of the strength of the spin-orbit coupling; it also fixes the value of the Raman recoil energy, $E_R = \hbar^2 k_R^2 / 2m$. The Raman coupling is quantified by $\Omega_R$. In writing Eq. (1) we have omitted a Zeeman shift term $\hbar \delta_R \sigma_z / 2$, where $\delta_R$ is the detuning of the Raman light field from resonance. The latter can be set equal to zero by properly choosing the frequency detuning $\Delta \omega_L$ of the Raman lasers. Equation (1) actually represents the Hamiltonian of the system in a spin-rotated frame, that is connected to the laboratory frame by the space- and time-dependent unitary transformation [49]

$$\mathcal{U} = \exp\left[ i(2k_R x - \Delta \omega_L t) \frac{\sigma_z}{2} \right]. \tag{2}$$

Hamiltonian (1) enjoys translational invariance as it commutes with the canonical momentum $\mathbf{p}$. Hence, one can look for eigenstates in the form of plane waves. The energy spectrum includes two branches, a lower and an upper one. For $\hbar \Omega_R < 4E_R$ the lower branch exhibits two degenerate minima at $\mathbf{p} = \pm \hbar k_1^{\text{SP}} \mathbf{e}_x$, where $\mathbf{e}_x$ denotes the unit vector along the $x$ direction and

$$k_1^{\text{SP}} = k_R \sqrt{1 - \left( \frac{\hbar \Omega_R}{4E_R} \right)^2} \,. \tag{3}$$

Instead, when $\hbar \Omega_R \geq 4E_R$ one has a single minimum located at $\mathbf{p} = 0$.

## 2.2 Many-body physics in the mean-field approach

In the weakly interacting regime a spin-orbit-coupled BEC, with the single-particle Hamiltonian of the form (1), can be safely described by mean-field theory. Indeed, the addition of the spin-orbit coupling has limited effects on the quantum depletion of a three-dimensional condensate [50, 51]; we checked that, for the set of parameters used in the present work, the quantum depletion remains on the order of a few percent. In the mean-field analysis the state of the system is represented by a two-component (spinor) wave function

$$\Psi(\mathbf{r}, t) = \begin{pmatrix} \Psi_\uparrow(\mathbf{r}, t) \\ \Psi_\downarrow(\mathbf{r}, t) \end{pmatrix}, \tag{4}$$

normalized to the total number of particles according to

$$\int_V d^3 r \, \Psi^\dagger \Psi = N \,, \tag{5}$$

where $V$ is the volume enclosing the atoms. We point out that, unlike $N$, the numbers of atoms in the up and down component, given by $N_{\uparrow, \downarrow} = \int_V d^3 r \, |\Psi_{\uparrow, \downarrow}|^2$, are not separately conserved because of the Raman coupling term in the spin-orbit Hamiltonian (1). The order parameter $\Psi(\mathbf{r}, t)$ relative to the ground-state many-body Bose-Einstein–condensed configuration evolves in time according to $\Psi(\mathbf{r}, t) = \Psi_0(\mathbf{r}) e^{-i\mu t / \hbar}$, with $\mu$ the chemical potential [44]. The total energy, including both single-particle and interaction terms, reads

$$E = \int_V d^3 r \left( \Psi^\dagger h_{\text{SO}} \Psi + \frac{g_{dd} n^2}{2} + \frac{g_{ss} s_z^2}{2} \right). \tag{6}$$

Here $n = \Psi^\dagger \Psi$ and $s_z = \Psi^\dagger \sigma_z \Psi$ are the total and spin density, respectively; notice that these quantities are unaffected by the unitary transformation (2) and take the same value in both the laboratory and the spin-rotated frame. In writing Eq. (6) we have assumed that the atoms in the BEC interact via a two-body contact potential. The density-density and spin-spin coupling constants are given by $g_{dd} = (g + g_{\uparrow\downarrow})/2$ and $g_{ss} = (g - g_{\uparrow\downarrow})/2$, respectively. In the following we will always assume equal intraspecies couplings, i.e., $g_{\uparrow\uparrow} = g_{\downarrow\downarrow} \equiv g$, together with the conditions $g > 0$ and $g_{dd} > 0$ ensuring stability against collapse. In the more general case (not addressed in this work) of asymmetric intraspecies couplings one should replace $g$ with $(g_{\uparrow\uparrow} + g_{\downarrow\downarrow})/2$ and add a term $g_{ds} n s_z$, with $g_{ds} = (g_{\uparrow\uparrow} - g_{\downarrow\downarrow})/4$, inside the integral of Eq. (6). The couplings in each channel are related to the corresponding $s$-wave scattering lengths via $g_{\sigma\sigma'} = 4\pi\hbar^2 a_{\sigma\sigma'}/m$ $(\sigma, \sigma' =\uparrow, \downarrow)$.

The zero-temperature phase diagram of the spin-orbit-coupled BEC under consideration was first investigated in Refs. [52, 53]. In these papers an Ansatz for the ground-state wave function $\Psi_0$ was introduced. This Ansatz consists of a linear combination of two plane waves with opposite momenta along $x$; each plane wave is multiplied by a two-component spinor wave function. All the quantities characterizing the order parameter $\Psi_0$ (the wave vectors of the two plane waves, their relative weights, and the components of the corresponding spinor wave functions) are determined by a procedure of minimization of the energy (6). The results of this variational procedure depend in a nontrivial way on the value of the average density of the gas $\bar{n} = N/V$. At relatively low densities and for $g_{ss} > 0$ the ground state of the BEC is compatible with three distinct quantum phases:

1. At low values of the Raman coupling $\Omega_R$ the system is in the stripe phase. In this phase the two momentum components in the wave function have equal weights, yielding a vanishing value of the spin polarization $\langle \sigma_z \rangle = \int_V d^3 r\, s_z$ along the $z$ axis. When calculating the condensate density, these two components interfere, giving rise to a periodically modulated density profile along the $x$ direction. The appearance of density modulations signals the spontaneous breaking of translational symmetry, occurring simultaneously with the spontaneous breaking of U(1) invariance. These features are typical of supersolid configurations. For this reason we will often refer to the stripe phase as to the supersolid phase.

2. At intermediate $\Omega_R$ the system enters the plane-wave phase, where the atoms condense in a state with finite momentum. Unlike the stripe phase, in the plane-wave phase the spin polarization $\langle \sigma_z \rangle$ takes a nonzero value.

3. Finally, at large $\Omega_R$ the gas is in the zero-momentum phase, where both the condensation momentum and the spin polarization along $z$ vanish.

The transition between the stripe and the plane-wave phase has a first-order nature. In the limit of vanishingly small $\bar{n}$ it occurs at a critical value of the Raman coupling given by [52]

$$\hbar\Omega_{\text{ST-PW}} = 4E_R \sqrt{\frac{2g_{ss}}{g_{dd} + 2g_{ss}}}\,. \tag{7}$$

At finite average density the value of $\Omega_{\text{ST-PW}}$ can be computed imposing that the chemical potential and pressure be equal in the two phases. One finds that at the transition such phases are at equilibrium with different values of their densities. However, the density differences are typically extremely small [49]. Hence, in the present work, and particularly in Figs. 1, 2, 3, 4, and 7, we estimate $\Omega_{\text{ST-PW}}$ without taking this effect into account and using the density as a fixed thermodynamic parameter. Another feature that is visible in the above figures is that, because of the first-order nature of the transition, the stripe phase can be found as a

metastable configuration even above the critical Raman coupling $\Omega_{ST-PW}$, and up to the so-called spinodal point. The situation is different for the transition between the plane-wave and zero-momentum phases, which has a second-order nature and takes place at the critical coupling [53]

$$\hbar\Omega_{PW-ZM} = 2(2E_R - g_{ss}\bar{n}). \tag{8}$$

The plane-wave and zero-momentum phases, as well as the corresponding phase transition at the critical point (8), also occur when $g_{ss} < 0$. Instead, the stripe phase does not exist under this condition, as it favors immiscible configurations [53].

We conclude by mentioning that for $g_{ss} > 0$ in the large-density regime the plane-wave phase is no longer energetically favored, and one has a direct first-order transition from the stripe to the zero-momentum phase [53, 54].

## 2.3 The stripe phase. General remarks

In the rest of this paper we will focus almost exclusively on the stripe phase. As mentioned in the previous section, in the analysis of Refs. [52, 53] the condensate wave function in this phase was taken as a superposition of two plane waves with opposite momenta. However, this is only an approximation to the exact wave function, that we write in the form [36]

$$\Psi_0(\mathbf{r}) = \sqrt{\bar{n}} \sum_{\bar{m}} \tilde{\Psi}_{\bar{m}} e^{i\bar{m}k_1 x}, \tag{9}$$

with $\bar{m} = \pm 1, \pm 3, \pm 5, \dots$. Here the spinor expansion coefficients $\tilde{\Psi}_{\bar{m}} = \begin{pmatrix} \tilde{\Psi}_{\bar{m},\uparrow} & \tilde{\Psi}_{\bar{m},\downarrow} \end{pmatrix}^T$ obey the condition $\sum_{\bar{m}} \tilde{\Psi}_{\bar{m}}^\dagger \tilde{\Psi}_{\bar{m}} = 1$, which ensures the correct normalization of $\Psi_0$. They also enjoy the symmetry property $\tilde{\Psi}_{-\bar{m}} = \sigma_x \tilde{\Psi}_{\bar{m}}^*$,[1] which implies the vanishing of the spin polarization $\langle \sigma_z \rangle$. Notice that the expansion (9) contains an infinite number of harmonic terms, each oscillating at an odd multiple wave vector of $k_1$, this last being a parameter to be adjusted to find the ground state. The variational Ansatz employed in Refs. [52, 53] can be recovered by truncating the expansion to the two terms with $\bar{m} = \pm 1$. The presence of higher-order harmonics is due to the nonlinear terms in the Gross-Pitaevskii equation [see Eq. (12)].

The order parameter (9) can be put in the form of a Bloch wave, i.e., a plane wave with quasimomentum $k_1$ times a periodic function with the same periodicity of the stripes. One can easily see that the wave vector of the density modulations corresponds to the wave-vector difference $2k_1$ between any two consecutive terms. As we will discuss below, it differs from the momentum difference of the two single-particle minima, given by $2k_1^{SP}$ [see Eq. (3)], because of interaction effects. These features are also reflected in the periodicity properties of the structure (9). It is antiperiodic in $x$ with antiperiod $\pi/k_1$, that is, $\Psi_0(x + \pi/k_1, y, z) = -\Psi_0(x, y, z)$, and thus it is also periodic with period $2\pi/k_1$.

One can prove that $k_1$ is related to the $\tilde{\Psi}_{\bar{m}}$'s in a simple way. This can be done inserting the Ansatz (9) into the energy (6). After performing the integration over the spatial coordinates one finds that $k_1$ explicitly appears only in the kinetic energy term along $x$,

$$
\begin{aligned}
E_{kin,x} &= \int_V d^3r \, \Psi_0^\dagger \frac{(p_x - \hbar k_R \sigma_z)^2}{2m} \Psi_0 \\
&= N \frac{\hbar^2}{2m} \sum_{\bar{m}} \tilde{\Psi}_{\bar{m}}^\dagger (\bar{m}k_1 - k_R \sigma_z)^2 \tilde{\Psi}_{\bar{m}}.
\end{aligned}
\tag{10}
$$

---

[1]This equality actually holds up to a phase factor, which can be taken equal to 1 by properly choosing the global phase of $\Psi_0$ and the origin of the $x$ axis.

Since $\Psi_0$ is the ground state of the system, the condition of stationarity $\frac{\partial E}{\partial k_1} = \frac{\partial E_{\text{kin},x}}{\partial k_1} = 0$ must hold, yielding

$$k_1 = k_R \frac{\sum_{\bar{m}} \bar{m} \tilde{\Psi}_{\bar{m}}^\dagger \sigma_z \tilde{\Psi}_{\bar{m}}}{\sum_{\bar{m}} \bar{m}^2 \tilde{\Psi}_{\bar{m}}^\dagger \tilde{\Psi}_{\bar{m}}} \,. \tag{11}$$

This expression reproduces the result of Ref. [53] if one truncates Eq. (9) to the two terms with $\bar{m} = \pm 1$; for similar reasons it reduces to Eq. (3) in the absence of interaction [see Eq. (21) below].

The expansion coefficients $\tilde{\Psi}_{\bar{m}}$ can be determined from energy minimization. However, in this work we shall adopt a different strategy, based on the Gross-Pitaevskii equation $i\hbar \frac{\partial \Psi}{\partial t} = \frac{\delta E}{\delta \Psi^\dagger}$. For a stationary state this equation takes the well-known time-independent form

$$\left\{ \frac{\hbar^2}{2m} \left[ (k_1 p_x - k_R \sigma_z)^2 + k_R^2 p_\perp^2 \right] + \frac{\hbar \Omega_R}{2} \sigma_x + g_{dd}(\Psi_0^\dagger \Psi_0) + g_{ss}(\Psi_0^\dagger \sigma_z \Psi_0)\sigma_z \right\} \Psi_0 = \mu \Psi_0 \,. \tag{12}$$

In the above expression we have introduced dimensionless coordinates defined by the following scaling transformations

$$x \to \frac{x}{k_1}, \qquad y \to \frac{y}{k_R}, \qquad z \to \frac{z}{k_R}, \tag{13a}$$

$$p_x \to \hbar k_1 p_x, \qquad p_y \to \hbar k_R p_y, \qquad p_z \to \hbar k_R p_z \,. \tag{13b}$$

The main advantage of this choice is that $\Psi_0$ as a function of the scaled $x$ coordinate has period $2\pi$; this will prove useful when dealing with the $\Omega_R$-dependence of $k_1$. For consistency we have to scale also the system volume, $V \to V/(k_1 k_R^2)$ (with the consequent scaling of the density $\bar{n}$ and the wave function $\Psi_0$), and the interaction strengths, $g_{dd,ss} \to k_1 k_R^2 g_{dd,ss}$. Notice that the products $G_{dd} = \bar{n} g_{dd}$ and $G_{ss} = \bar{n} g_{ss}$ remain unaffected by these scaling transformations.

Equation (12) represents the starting point of the analysis of the next section. In particular, we will develop a perturbation approach to solve it in the limit of small values of the Raman coupling.

## 3 Perturbation analysis of the ground state

The present section is devoted to a perturbative study of the ground state of a spin-orbit-coupled BEC in the stripe phase. We first provide the exact solution in the limit of zero Raman coupling (Sec. 3.1). Then, in Sec. 3.2 we introduce the perturbation approach and present the results for several observables up to second order in the Raman coupling. The explicit calculation of the ground-state wave function is illustrated in Appendix A.

### 3.1 Ground state at $\Omega_R = 0$

In the absence of Raman coupling ($\Omega_R = 0$) the order parameter minimizing the spin-orbit energy functional (6) takes the form

$$\Psi_0^{(0)}(\mathbf{r}) = \sqrt{\bar{n}} \left[ \tilde{\Psi}_{+1}^{(0)} e^{ix} + \tilde{\Psi}_{-1}^{(0)} e^{-ix} \right], \tag{14}$$

with $\tilde{\Psi}_{+1}^{(0)} = e^{-i\chi_0/2} \begin{pmatrix} 1 & 0 \end{pmatrix}^T / \sqrt{2}$, $\tilde{\Psi}_{-1}^{(0)} = e^{i\chi_0/2} \begin{pmatrix} 0 & 1 \end{pmatrix}^T / \sqrt{2}$, $\chi_0$ being an arbitrary phase, and $k_1^{(0)} = k_R$. This configuration corresponds to the ground state of an interacting quantum mixture with $N_\uparrow = N_\downarrow$, equal intraspecies interactions, and $g_{ss} > 0$. The plane-wave dependence of the wave function, fixed by the value of $k_R$, is the consequence of the unitary transformation (2) employed to write the spin-orbit Hamiltonian in the spin-rotated frame [see Eq. (1)].

We stress that the state described by the wave function (14) has not a supersolid character. Actually, this configuration has uniform density ($n^{(0)} = \bar{n}$), as a consequence of the orthogonality of the two spinors $\tilde{\Psi}_{+1}^{(0)}$ and $\tilde{\Psi}_{-1}^{(0)}$. It is furthermore characterized by the vanishing of the spin density ($s_z^{(0)} = 0$), by the value $E^{(0)}/N = G_{dd}/2$ of the total energy per particle, following from Eq. (6), and by the chemical potential $\mu^{(0)} = G_{dd}$.

The physical meaning of the relative phase $\chi_0$ between the two spin components can be better understood by reverting to the laboratory frame. In this frame the spin polarization vector $(\langle \sigma_x \rangle, \langle \sigma_y \rangle, \langle \sigma_z \rangle) = N (\cos(\Delta\omega_L t + \chi_0), \sin(\Delta\omega_L t + \chi_0), 0)$ lies in the $(x, y)$ plane, its direction at a given time $t$ being fixed by $\chi_0$; hence, this miscible configuration can be regarded as an easy-plane ferromagnet. The arbitrariness of the value of $\chi_0$ reveals that spin-rotation invariance about the $z$ axis is spontaneously broken in the miscible phase. Notice that these considerations no longer hold at finite $\Omega_R$, as the Raman coupling term in the Hamiltonian (1) explicitly breaks spin-rotation symmetry.

## 3.2 Perturbation results for the ground state

In the following we will consider the $\hbar\Omega_R/4E_R \ll 1$ case, corresponding to the deep double-minimum regime of the single-particle spectrum. In this regime the Raman coupling term can be treated as a perturbation. The procedure used for determining $\Psi_0$ is very similar to that of Ref. [55], which considered a single-component BEC in a one-dimensional optical lattice. We perturbatively solve Eq. (12) expanding the wave function around the unperturbed configuration (14):

$$\Psi_0 = \Psi_0^{(0)} + \sum_{n=1}^{+\infty} \Psi_0^{(n)}. \tag{15}$$

Here the superscript "$(n)$" denotes the correction of order $(\hbar\Omega_R/4E_R)^n$ to the wave function of the condensate. Inserting the expansion (15), as well the corresponding expansions

$$k_1 = k_R + \sum_{n=1}^{+\infty} k_1^{(n)} \tag{16}$$

for $k_1$ and

$$\mu = G_{dd} + \sum_{n=1}^{+\infty} \mu^{(n)} \tag{17}$$

for the chemical potential, into Eq. (12), and equating terms of the same order on both sides, one finds recurrence relations that can be solved at each order. The explicit form of these recurrence relations [which take explicitly into account the normalization of the wave function and the constraints imposed by the U(1) and translational symmetries of the model], as well as of their solutions up to second order in $\hbar\Omega_R/4E_R$, is reported in Appendix A.

Here we provide the relevant results for the expansion of the physical quantities of major interest. In particular the contrast of stripes, defined by $\mathcal{C} = (n_{\max} - n_{\min})/(n_{\max} + n_{\min})$ with $n_{\min}$ ($n_{\max}$) the minimum (maximum) value taken by the density modulations as a function of $x$, is found to obey the linear dependence

$$\mathcal{C} = \frac{2E_R}{2E_R + G_{dd}} \frac{\hbar\Omega_R}{4E_R} \tag{18}$$

on $\hbar\Omega_R/4E_R$, higher-order corrections being of order $(\hbar\Omega_R/4E_R)^3$. In general, $\mathcal{C}$ is antisymmetric under $\Omega_R \to -\Omega_R$ because this operation exchanges the locations of the maxima and minima of the density modulations. It is worth mentioning that the actual position of the fringes is the result of a spontaneous breaking mechanism of translational invariance. In the present

formalism it is fixed by the value of the relative phase $\chi_0$ between the two components of the $\Omega_R = 0$ wave function (14) and by the condition (68b) imposed on the perturbative corrections (see the related discussion in Appendix A.1).

Other physical quantities, like the chemical potential and the energy per particle, as well as the wave vector $k_1$ fixing the periodicity of the density modulations have instead vanishing first-order contributions, the lowest corrections to the unperturbed values being quadratic in $\hbar\Omega_R/4E_R$. The perturbation approach provides the following results:

$$\mu = G_{dd} - \frac{E_R(8E_R^2 + 4G_{dd}E_R + G_{dd}^2)}{2(2E_R + G_{dd})^2} \left(\frac{\hbar\Omega_R}{4E_R}\right)^2 , \tag{19}$$

$$\frac{E}{N} = \frac{G_{dd}}{2} - \frac{E_R(4E_R + G_{dd})}{2(2E_R + G_{dd})} \left(\frac{\hbar\Omega_R}{4E_R}\right)^2 , \tag{20}$$

$$k_1 = k_R - k_R \frac{4E_R^2 + 2G_{dd}E_R + G_{dd}^2}{2(2E_R + G_{dd})^2} \left(\frac{\hbar\Omega_R}{4E_R}\right)^2 . \tag{21}$$

The above perturbation results can be compared with the exact values, obtained by numerically minimizing the energy (6) for a wave function of the form (9). Such a comparison is presented in Fig. 1, where the same interaction parameters as Ref. [36] have been taken. Notice that for all the considered quantities the agreement is excellent within a wide range of values of $\Omega_R$, extending up to values close to the transition to the plane-wave phase. This confirms the validity and usefulness of the perturbation approach developed in the present work. The above choice of the interaction parameters actually amplifies the range of values of $\Omega_R$ characterizing the stripe-phase region, yielding a significant increase of the critical Raman coupling for the transition to the plane-wave phase, which occurs at $\hbar\Omega_R = 2.70\, E_R$ (even though the stripe phase continues to exist as a metastable configuration up to the spinodal point at $\hbar\Omega_R = 2.85\, E_R$). Unfortunately these values of the parameters do not correspond to the current values of the scattering lengths of alkali atoms. In particular, because of the smallness of the ratio $g_{ss}/g_{dd} \approx 10^{-3}$ in $^{87}$Rb the transition to the plane-wave phase takes place at a much lower critical coupling, whose value $\hbar\Omega_R = 0.19\, E_R$ was measured in [48] and found to be in excellent agreement with the theoretical prediction (7). Consequently, the emerging supersolid effects, like the contrast of fringes, are very small in this case. However, a decrease of the effective interspecies coupling constant $g_{\uparrow\downarrow}$ (and hence an increase of the critical value of $\Omega_R$) can be achieved by reducing the spatial overlap between the wave functions of the two spin [56] or pseudo-spin [57] components. This strategy, which already led to the first successful detection of the stripe phase employing the pseudo-spin states of a superlattice [10], could open realistic perspectives for the observability of the sizable effects predicted in the present work. Another promising strategy is provided by the use of different atomic species, like $^{39}$K, where the availability of Feshbach resonances [58] is opening novel perspectives to increase the critical value of the Raman coupling giving the transition to the plane-wave phase [45].

## 4 Bogoliubov modes in the stripe phase

The perturbation approach developed in the previous section can be naturally extended to the study of the Bogoliubov modes of our striped condensate. As a preliminary step, in Sec. 4.1 we briefly review the fundamentals of Bogoliubov theory and of its formulation in the stripe phase. In Sec. 4.2 we summarize the properties of the Bogoliubov modes in the absence of Raman coupling while in Sec. 4.3 we discuss the results for the most relevant features of the Bogoliubov solutions holding up to first and second order in the perturbation parameter $\hbar\Omega_R/4E_R$. Special focus is given on the Goldstone modes exhibited by the stripe phase. Subsequently we

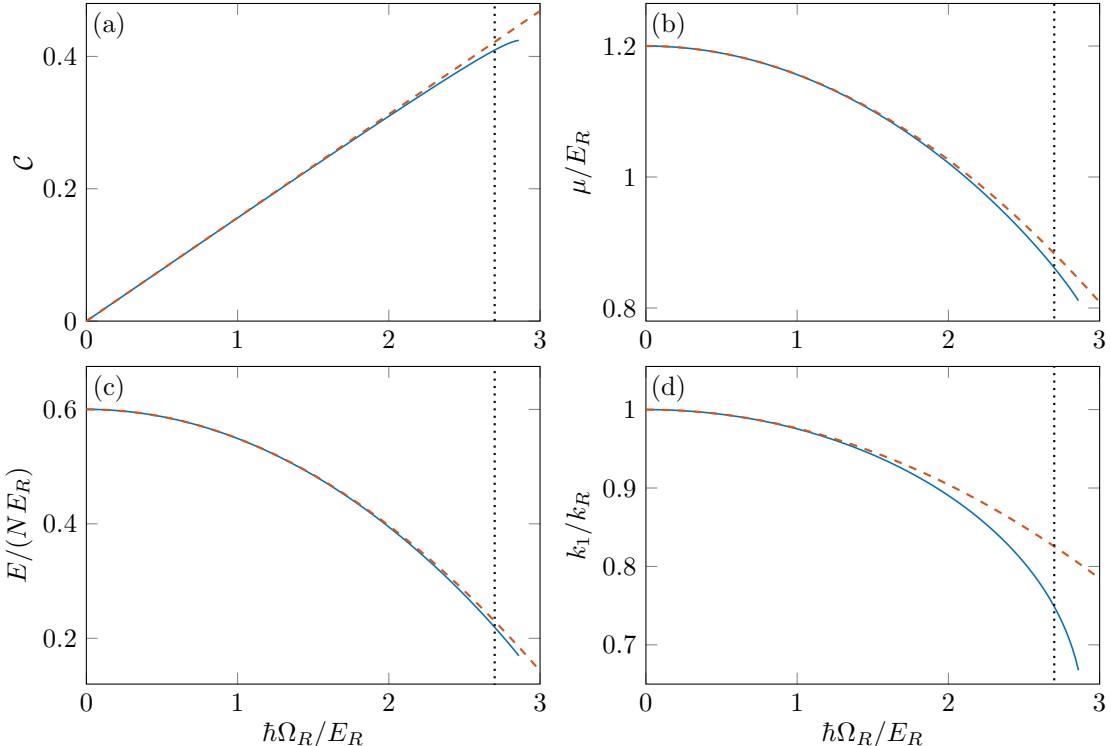

Figure 1: (a) contrast, (b) chemical potential, (c) energy per particle, and (d) wave vector in the stripe phase as functions of the Raman coupling $\Omega_R$ for a fixed value of the density. The blue solid lines show the exact values determined numerically, whereas the red dashed ones correspond to the perturbation results up to second order in $\hbar\Omega_R/4E_R$. The interaction parameters are the same as in Ref. [36]: $G_{dd}/E_R = 1.2$, $G_{ss}/E_R = 0.32$. As stated in the main text, for such values the first-order transition to the plane-wave phase occurs at $\hbar\Omega_R = 2.70\,E_R$ (vertical black dotted lines), and the spinodal point at $\hbar\Omega_R = 2.85\,E_R$.

address the computation of the sum rules in the long-wavelength limit (Sec. 4.4) and of the superfluid density (Sec. 4.5). In all these sections we systematically compare the perturbation approach with the exact solution of the Bogoliubov equations. The detailed perturbation calculation of the Bogoliubov modes is illustrated in Appendix B.

## 4.1 Basics of Bogoliubov theory. Results in the stripe phase

The starting point of our analysis is the time-dependent Gross-Pitaevskii equation

$$i\hbar\partial_t\Psi = \left[h_{\mathrm{SO}} + g_{dd}\left(\Psi^\dagger\Psi\right) + g_{ss}\left(\Psi^\dagger\sigma_z\Psi\right)\sigma_z\right]\Psi, \tag{22}$$

whose linearized solutions, that we write in the form

$$\Psi(\mathbf{r}, t) = e^{-i\mu t/\hbar}\left[\Psi_0(\mathbf{r}) + \delta\Psi(\mathbf{r}, t)\right], \tag{23}$$

are known to be equivalent to the formulation of Bogoliubov theory [43, 44, 59]. In the above equation $\Psi_0(\mathbf{r})$ is the equilibrium configuration studied in the previous section, while the fluctuation term $\delta\Psi$ obeys the linearized Bogoliubov equation

$$i\hbar\partial_t\delta\Psi = (h_{\mathrm{SO}} - \mu + h_{\mathrm{D}})\,\delta\Psi + h_{\mathrm{C}}\delta\Psi^*. \tag{24}$$

Here we have introduced the two matrices

$$h_{\mathrm{D}} = g_{dd}(\Psi_0^{\dagger}\Psi_0\mathbb{I}_2 + \Psi_0\Psi_0^{\dagger}) + g_{ss}[(\Psi_0^{\dagger}\sigma_z\Psi_0)\sigma_z + (\sigma_z\Psi_0)(\sigma_z\Psi_0)^{\dagger}], \tag{25a}$$

$$h_{\mathrm{C}} = g_{dd}\Psi_0\Psi_0^{T} + g_{ss}(\sigma_z\Psi_0)(\sigma_z\Psi_0)^{T}, \tag{25b}$$

where $\mathbb{I}_2$ denotes the $2 \times 2$ identity matrix.

We look for solutions of Eq. (24) in the form

$$\delta\Psi(\mathbf{r}, t) = \lambda U(\mathbf{r})e^{-i\omega t} + \lambda^* V^*(\mathbf{r})e^{i\omega t}. \tag{26}$$

Here $\omega$ is the frequency at which the wave function oscillates around $\Psi_0$, with $U$ and $V$ the corresponding two-component (spinor) Bogoliubov amplitudes, and $\lambda$ is a complex multiplicative factor such that $|\lambda| \ll 1$. Inserting Eq. (26) into Eq. (24) one finds that the Bogoliubov frequencies and amplitudes are solutions of the eigenvalue problem

$$\eta\mathcal{B}\mathcal{W} = \hbar\omega\mathcal{W}. \tag{27}$$

Here, for notational compactness, we have defined the two $4 \times 4$ matrices

$$\eta = \begin{pmatrix} \mathbb{I}_2 & 0 \\ 0 & -\mathbb{I}_2 \end{pmatrix} \tag{28}$$

and

$$\mathcal{B} = \begin{pmatrix} h_{\mathrm{SO}} - \mu + h_{\mathrm{D}} & h_{\mathrm{C}} \\ h_{\mathrm{C}}^* & (h_{\mathrm{SO}} - \mu + h_{\mathrm{D}})^* \end{pmatrix}, \tag{29}$$

as well as the four-component Bogoliubov column vectors

$$\mathcal{W} = \begin{pmatrix} U \\ V \end{pmatrix}, \quad \bar{\mathcal{W}} = \begin{pmatrix} V^* \\ U^* \end{pmatrix}. \tag{30}$$

The column vector $\mathcal{W}$ is normalized to

$$\int_V d^3r\, \mathcal{W}^{\dagger}\eta\mathcal{W} = \int_V d^3r\, (U^{\dagger}U - V^{\dagger}V) = 1. \tag{31}$$

For further details concerning the Bogoliubov formalism applied to the spinor configuration see Appendix B.

If the condensate is in the stripe phase described by the mean-field wave function (9), the solutions of Eq. (27) can be expressed as Bloch waves of the form [36, 60]

$$U_{b,\mathbf{k}}(\mathbf{r}) = e^{i\mathbf{k}\cdot\mathbf{r}}\sum_{\bar{m}} \tilde{U}_{b,\mathbf{k},\bar{m}}e^{i\bar{m}x}, \tag{32a}$$

$$V_{b,\mathbf{k}}(\mathbf{r}) = e^{i\mathbf{k}\cdot\mathbf{r}}\sum_{\bar{m}} \tilde{V}_{b,\mathbf{k},\bar{m}}e^{i\bar{m}x}, \tag{32b}$$

with $\omega_{b,\mathbf{k}}$ the corresponding frequency. Here $\tilde{U}_{b,\mathbf{k},\bar{m}}$ and $\tilde{V}_{b,\mathbf{k},\bar{m}}$ are expansion coefficients, and $\mathbf{k}$ is a quasimomentum. We take the $x$ component of $\mathbf{k}$ in the first Brillouin zone, i.e., $0 \le k_x < 2$, while $k_y$ and $k_z$ are unbounded. Notice that, for consistency with Eq. (13), the scaling

$$k_x \to k_1 k_x, \quad k_y \to k_R k_y, \quad k_z \to k_R k_z \tag{33}$$

has been performed, such that $\mathbf{k}$ is dimensionless. For a fixed value of $\mathbf{k}$ one finds an infinite number of solutions of Eq. (27), which implies that the Bogoliubov spectrum has a band structure; we use the additional index $b = 1, 2, \ldots$ to distinguish between the various bands (see

also [36,60]). Most importantly, the spectrum exhibits two gapless branches, whose frequency vanishes at the edge of the Brillouin zone (see Fig. 6 in Appendix B). The occurrence of these Goldstone modes is the consequence of the spontaneous breaking of U(1) and translational symmetry.

The Bogoliubov formalism is particularly useful for studying the dynamics of the system when a weak perturbation, of the form $V_{\text{pert}} = -\epsilon \mathcal{O} e^{-i\omega t} + \text{H.c.}$ with $|\epsilon| \ll 1$ and $\mathcal{O}$ a given operator, is added to the single-particle Hamiltonian (1). Typical choices for $\mathcal{O}$ are the $\mathbf{q}$-component of the density operator, $\rho_{\mathbf{q}} = e^{i\mathbf{q}\cdot\mathbf{r}}$, and of the spin-density operator, $s_{z,\mathbf{q}} = \sigma_z e^{i\mathbf{q}\cdot\mathbf{r}}$. A fundamental object to compute is the dynamic structure factor [44]

$$S_{\mathcal{O}}(\omega) = \sum_{b,\mathbf{k}} |\langle 0|\mathcal{O}|b,\mathbf{k}\rangle|^2 \, \delta(\hbar\omega - \hbar\omega_{b,\mathbf{k}}). \tag{34}$$

Here $\langle 0|\mathcal{O}|b,\mathbf{k}\rangle$ denotes the matrix element of $\mathcal{O}$ between the ground state and a given excited mode. Its square modulus is called the strength of $\mathcal{O}$ in the mode $(b,\mathbf{k})$; a simple calculation yields

$$|\langle 0|\mathcal{O}|b,\mathbf{k}\rangle|^2 = \left| \int_V d^3 r \left( \Psi_0^\dagger \mathcal{O} U_{b,\mathbf{k}} + V_{b,\mathbf{k}}^T \mathcal{O} \Psi_0 \right) \right|^2. \tag{35}$$

Notice that for the above-mentioned operators $\rho_{\mathbf{q}}$ and $s_{z,\mathbf{q}}$, as well as for the transverse current operator introduced in Eq. (58) below, the strength vanishes unless the difference $\mathbf{q} - \mathbf{k}$ equals an integer multiple of $2k_1 \mathbf{e}_x$.

We define the $p$-th moment of the dynamic structure factor as

$$
\begin{aligned}
m_p(\mathcal{O}) &= \hbar^{p+1} \int d\omega \, \omega^p S_{\mathcal{O}}(\omega) \\
&= \sum_{b,\mathbf{k}} (\hbar\omega_{b,\mathbf{k}})^p \, |\langle 0|\mathcal{O}|b,\mathbf{k}\rangle|^2 .
\end{aligned}
\tag{36}
$$

These moments are known to obey important sum rules [44]. For instance, the $p = 0$ moment gives the static structure factor $S(\mathcal{O}) = m_0(\mathcal{O})/N$. In this work we will make large use of the identity

$$\chi(\mathcal{O}) = N^{-1}[m_{-1}(\mathcal{O}) + m_{-1}(\mathcal{O}^\dagger)], \tag{37}$$

relating the $p = -1$ moment to the static response function $\chi(\mathcal{O})$.

In the following we implement a perturbation approach to study the Bogoliubov modes and analytically calculate several relevant quantities in the limit of small Raman coupling.

## 4.2 Bogoliubov spectrum at $\Omega_R = 0$

As discussed in Sec. 3.1, in the $\Omega_R \to 0$ limit the condensate wave function approaches that of a uniform unpolarized mixture. Furthermore, because of the vanishing of the Raman coupling term, the spin-orbit Hamiltonian (1) commutes with the physical momentum $\mathbf{q} = \mathbf{p} - \sigma_z \mathbf{e}_x$. Thus, we can label the excited states using the eigenvalues of the $\mathbf{q}$ operator. The Bogoliubov spectrum as a function of $\mathbf{q}$ is made of two branches, one featuring density oscillations, the other spin oscillations [43, 44], with frequencies

$$\hbar\omega_{d,\mathbf{q}}^{(0)} = \sqrt{\hbar\Omega_{\mathbf{q}} \left( \hbar\Omega_{\mathbf{q}} + 2G_{dd} \right)}, \tag{38a}$$

$$\hbar\omega_{s,\mathbf{q}}^{(0)} = \sqrt{\hbar\Omega_{\mathbf{q}} \left( \hbar\Omega_{\mathbf{q}} + 2G_{ss} \right)}, \tag{38b}$$

respectively. In the above equations $\hbar\Omega_{\mathbf{q}} = E_R q^2$ and we employed, for the components of $\mathbf{q}$, the same scaling transformations used for the quasimomentum [see Eq. (33)]. The corresponding Bogoliubov amplitudes are given by

$$\mathcal{W}_{d,\mathbf{q}}^{(0)}(\mathbf{r}) = \begin{pmatrix} U_{d,\mathbf{q}}^{(0)}(\mathbf{r}) \\ V_{d,\mathbf{q}}^{(0)}(\mathbf{r}) \end{pmatrix} = \frac{e^{i\mathbf{q}\cdot\mathbf{r}}}{\sqrt{N}} \begin{pmatrix} u_{d,\mathbf{q}}\Psi_0^{(0)}(\mathbf{r}) \\ v_{d,\mathbf{q}}[\Psi_0^{(0)}(\mathbf{r})]^* \end{pmatrix}, \tag{39a}$$

$$\mathcal{W}_{s,\mathbf{q}}^{(0)}(\mathbf{r}) = \begin{pmatrix} U_{s,\mathbf{q}}^{(0)}(\mathbf{r}) \\ V_{s,\mathbf{q}}^{(0)}(\mathbf{r}) \end{pmatrix} = \frac{e^{i\mathbf{q}\cdot\mathbf{r}}}{\sqrt{N}} \begin{pmatrix} u_{s,\mathbf{q}}\sigma_z\Psi_0^{(0)}(\mathbf{r}) \\ v_{s,\mathbf{q}}[\sigma_z\Psi_0^{(0)}(\mathbf{r})]^* \end{pmatrix}, \tag{39b}$$

where $\Psi_0^{(0)}$ is the ground-state wave function [see Eq. (14)], and

$$u_{\ell,\mathbf{q}} = \frac{1}{2}\left(\sqrt{\frac{\Omega_{\mathbf{q}}}{\omega_{\ell,\mathbf{q}}}} + \sqrt{\frac{\omega_{\ell,\mathbf{q}}}{\Omega_{\mathbf{q}}}}\right), \tag{40a}$$

$$v_{\ell,\mathbf{q}} = \frac{1}{2}\left(\sqrt{\frac{\Omega_{\mathbf{q}}}{\omega_{\ell,\mathbf{q}}}} - \sqrt{\frac{\omega_{\ell,\mathbf{q}}}{\Omega_{\mathbf{q}}}}\right), \tag{40b}$$

with $\ell = d, s$ the branch index. Notice that the dispersion laws (38) are isotropic, i.e., they do not depend on the direction of $\mathbf{q}$. In the low-$q$ limit they are linear in $q$, $\omega_{\ell,\mathbf{q}}^{(0)} \sim c_\ell^{(0)} q$, and the sound velocities for density and spin waves, using the unscaled momentum $\mathbf{q}$, are given by

$$c_d^{(0)} = \sqrt{\frac{G_{dd}}{m}}, \quad c_s^{(0)} = \sqrt{\frac{G_{ss}}{m}}. \tag{41}$$

As soon as $\Omega_R \neq 0$ the physical momentum $\mathbf{q}$ is no longer a good quantum number, and has to be replaced by the quasimomentum $\mathbf{k}$ (see Sec. 4.1). Actually one can define $\mathbf{k}$ also at vanishing $\Omega_R$. In this case the two original branches (38) of the Bogoliubov spectrum split into an infinite number of bands, each identified by the band index $b$ and defined in the first Brillouin zone, as discussed in Appendix B. As shown in Fig. 5, these bands can cross each other, and the crossing points (marked in black in the figure) correspond to Bogoliubov modes that are degenerate at $\Omega_R = 0$. This degeneracy is lifted as soon as the Raman coupling is turned on, leading to the opening of gaps between the excitation bands. For simplicity we will not deal with this mechanism, whose theoretical treatment requires the use of an involved degenerate perturbation method. On the other hand, we stress that the simpler nondegenerate perturbation approach developed in this work is well suited for studying the long-wavelength limit of the lowest-lying bands. This will be the core of the discussion of the next section.

### 4.3 Perturbation results for the Bogoliubov modes

We shall now perform a perturbation study of the Bogoliubov modes in the $\hbar\Omega_R/4E_R \ll 1$ limit, analogous to that of the ground state made in Sec. 3. We start by expanding the Bogoliubov frequencies and amplitudes,

$$\omega_{b,\mathbf{k}} = \omega_{b,\mathbf{k}}^{(0)} + \sum_{n=1}^{+\infty} \omega_{b,\mathbf{k}}^{(n)}, \quad \mathcal{W}_{b,\mathbf{k}} = \mathcal{W}_{b,\mathbf{k}}^{(0)} + \sum_{n=1}^{+\infty} \mathcal{W}_{b,\mathbf{k}}^{(n)}, \tag{42}$$

as well as the matrix $\mathcal{B}$ of Eq. (29),

$$\mathcal{B} = \mathcal{B}^{(0)} + \sum_{n=1}^{+\infty} \mathcal{B}^{(n)}, \tag{43}$$

where the scaling transformations (13) are implicitly assumed. The formalism of the expansion procedure is discussed in details in Appendix B, where we derive explicit expressions for the frequencies and the Bogoliubov amplitudes of the various bands up to quadratic terms in the perturbation parameter $\hbar\Omega_R/4E_R$.

Here we provide explicit results for the most interesting lowest-energy branches in the long-wavelength limit (corresponding to $\ell = s, d$, $\bar{K} = 0$, and $k \to 0$ in the notation of Appendix B), where they reveal their Goldstone nature. These modes, which are characterized by both small quasimomentum and small frequency, have a phonon-like dispersion law, $\omega_{\ell,0,\mathbf{k}} \sim c_\ell(\theta_\mathbf{k})k$. Here $c_\ell$ is the sound velocity in the branch $\ell$; due to the anisotropy of our system, which is caused by the spin-orbit coupling, it generally depends on the angle $\theta_\mathbf{k}$ between $\mathbf{k}$ and the positive $x$ direction [36, 49] (see Appendix B).

The second-order expansion of the sound velocities relative to the two bands, calculated along the $x$ direction ($\theta_\mathbf{k} = 0$), yields the results

$$c_{d,x} = c_d^{(0)} - \frac{\left[ G_{dd}^3 E_R + 6G_{dd}^2 E_R^2 + 2G_{dd}E_R^2(8E_R + G_{ss}) + 8E_R^4 \right] c_d^{(0)}}{2(2E_R + G_{dd})^3(2E_R + G_{ss})} \left( \frac{\hbar\Omega_R}{4E_R} \right)^2 , \tag{44a}$$

$$\begin{aligned}
c_{s,x} = c_s^{(0)} &- \frac{c_s^{(0)}}{2G_{ss}(2E_R + G_{dd})^3(2E_R + G_{ss})} \\
&\times \Big[ G_{dd}^3 \left( 2E_R^2 + 5G_{ss}E_R + 2G_{ss}^2 \right) + G_{dd}^2 E_R \left( 8E_R^2 + 30G_{ss}E_R + 13G_{ss}^2 \right) \\
&+ G_{dd}E_R \left( 8E_R^3 + 40G_{ss}E_R^2 + 22G_{ss}^2 E_R + G_{ss}^3 \right) \\
&+ 2G_{ss}E_R^2 \left( 16E_R^2 + 12G_{ss}E_R + G_{ss}^2 \right) \Big] \left( \frac{\hbar\Omega_R}{4E_R} \right)^2 .
\end{aligned} \tag{44b}$$

The expansion of the sound velocities calculated along the directions perpendicular to $x$ ($\theta_\mathbf{k} = \pi/2$) instead gives

$$c_{d,\perp} = c_d^{(0)} + \frac{2E_R^3 c_d^{(0)}}{(2E_R + G_{dd})^3} \left( \frac{\hbar\Omega_R}{4E_R} \right)^2 , \tag{45a}$$

$$\begin{aligned}
c_{s,\perp} = c_s^{(0)} &- \frac{c_s^{(0)}}{2G_{ss}(2E_R + G_{dd})^2(2E_R + G_{ss})} \\
&\times \Big[ G_{dd}^2 E_R (2E_R + G_{ss}) + 2G_{dd}E_R \left( 2E_R^2 + 5G_{ss}E_R + 2G_{ss}^2 \right) \\
&+ G_{ss}E_R \left( 8E_R^2 + 8G_{ss}E_R + G_{ss}^2 \right) \Big] \left( \frac{\hbar\Omega_R}{4E_R} \right)^2 .
\end{aligned} \tag{45b}$$

The velocities of phonons propagating parallel and perpendicular to the $x$ axis are plotted in the upper and lower panel of Fig. 2, respectively. Remarkably, the perturbative estimates (dashed lines) turn out to be very close to the exact numerical solutions of the Bogoliubov equations (solid lines) for a wide range of values of the Raman coupling. Equations (44) and (45) show that the dependence of the sound velocities on the interaction parameters is considerably more involved than in binary mixtures without spin-orbit coupling. Notice that, although in the figures of this paper we take $G_{dd} > G_{ss}$, our analytical formulas also hold in the opposite case. An important feature of the sound velocities plotted in Fig. 2 is that they remain positive even beyond the transition to the plane-wave phase. More generally, the stripe phase is dynamically and energetically stable (i.e., the excitation spectrum is real and positive) up to the spinodal point as a consequence of its metastability, see Sec. 2.2. As shown in Fig. 2(a), the velocity of spin waves along $x$ vanishes at the spinodal point, where the stripe phase becomes dynamically unstable; this effect, which is associated with the divergence of the magnetic susceptibility (see Sec. 4.4), in trapped systems manifests itself in the softening of the spin-dipole frequency as the spinodal point is approached [45]. We point out that in the

$(\Omega_R, g_{ss})$ plane the transition from the stripe to the plane-wave phase continuously connects to the miscible-immiscible transition, occurring at $\Omega_R = 0$ when $g_{ss}$ is tuned from positive to negative values. However, no metastability window exists across the miscible-immiscible phase transition, as the miscible phase becomes dynamically unstable precisely at the critical point $g_{ss} = 0$, following the vanishing of the spin sound velocity $c_s^{(0)} = \sqrt{G_{ss}/m}$.

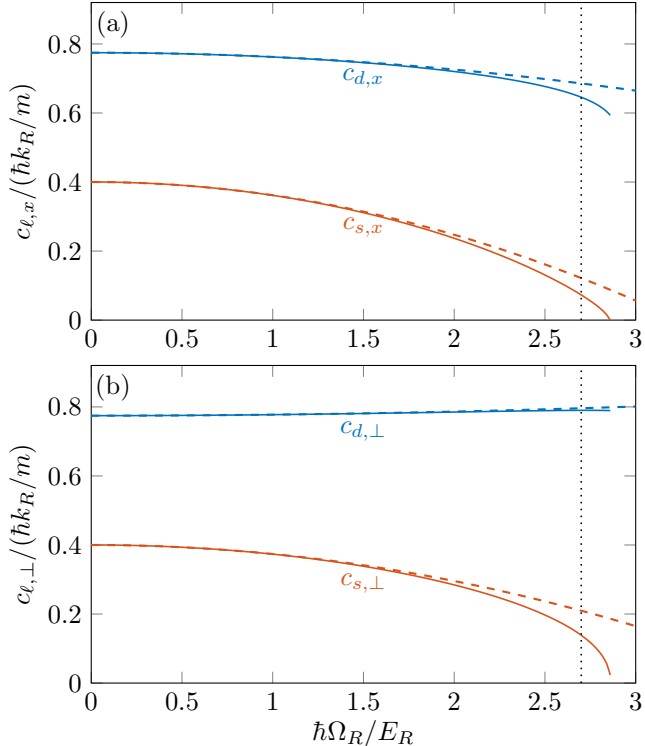

Figure 2: Sound velocity as a function of the Raman coupling $\Omega_R$. The results for excitations propagating parallel (perpendicular) to the $x$ axis are shown in panel (a) [(b)]. The solid curves represent the exact numerical values, the dashed ones correspond to the perturbation results obtained for the density (blue) and spin (red) gapless mode. The vertical dotted lines mark the critical Raman coupling $\hbar\Omega_R = 2.70\,E_R$ at which the transition to the plane-wave phase takes place. The interaction parameters are the same as Fig. 1.

We next look at the low-$k$ behavior of the Bogoliubov amplitudes of the two gapless bands, which can be related to the ground-state wave function $\Psi_0$ as follows:

$$\mathcal{W}_{d,0,\mathbf{k}} = \begin{pmatrix} U_{d,0,\mathbf{k}} \\ V_{d,0,\mathbf{k}} \end{pmatrix} \underset{k\to 0}{\sim} \frac{1}{2}\sqrt[4]{\frac{4mG_{dd}}{\hbar^2 k^2 N^2}} \begin{pmatrix} \Psi_0 \\ -\Psi_0^* \end{pmatrix}, \tag{46a}$$

$$\mathcal{W}_{s,0,\mathbf{k}} = \begin{pmatrix} U_{s,0,\mathbf{k}} \\ V_{s,0,\mathbf{k}} \end{pmatrix} \underset{k\to 0}{\sim} \frac{1}{2ik_R}\sqrt[4]{\frac{4mG_{ss}}{\hbar^2 k^2 N^2}} \begin{pmatrix} \nabla_x \Psi_0 \\ \nabla_x \Psi_0^* \end{pmatrix} \tag{46b}$$

(in this section and in the next two ones we use dimensional coordinates and momenta), where the dependence on the Raman coupling $\Omega_R$ is implicitly contained in $\Psi_0$. One can immediately see that Eqs. (46) reduce to results (39) at zero Raman coupling, as a consequence of the identity

$$\nabla_x \Psi_0^{(0)} = ik_R \sigma_z \Psi_0^{(0)}. \tag{47}$$

Equations (46) have been evaluated including linear terms in $\hbar\Omega_R/4E_R$. More general results accounting for higher-order corrections are presented in Appendix B.4. The full condensate

wave function (23)-(26) for these two modes, including both the equilibrium and the fluctuation terms, is readily evaluated,

$$\Psi_{d,0,\mathbf{k}}(\mathbf{r},t) \underset{k\to 0}{\sim} e^{-i\mu t/\hbar}(1+i\delta\varphi)\Psi_0(\mathbf{r}),\tag{48a}$$

$$\Psi_{s,0,\mathbf{k}}(\mathbf{r},t) \underset{k\to 0}{\sim} e^{-i\mu t/\hbar}(1+\delta x\,\nabla_x)\Psi_0(\mathbf{r}).\tag{48b}$$

The previous equations give direct insight on the physical nature of the two phonon modes in the macroscopic limit $k\to 0$. The former result represent an infinitesimal U(1) transformation of the wave function, that changes its phase by $\delta\varphi = \mathrm{Im}\,\lambda\sqrt[4]{4mG_{dd}/\hbar^2k^2N^2}$, where $\lambda$ accounts for the size of the fluctuation term $\delta\Psi$ [see Eq. (26)]; this is the standard Goldstone mode associated with the superfluid character of a Bose-Einstein condensate, featuring, to leading order, a simple change of the phase. Equation (48b) instead corresponds to an infinitesimal translation of the wave function by a displacement $\delta x = \mathrm{Im}\,\lambda\sqrt[4]{4mG_{ss}/\hbar^2k^2N^2}k_R^{-1}$ along $x$; in particular the total density

$$\Psi_{s,0,\mathbf{k}}^\dagger(\mathbf{r},t)\Psi_{s,0,\mathbf{k}}(\mathbf{r},t) \underset{k\to 0}{\sim} n(x+\delta x,y,z)\tag{49}$$

is characterized by shifted fringes compared to the equilibrium profile $n(\mathbf{r})$. This is the crystal Goldstone mode that typically appears in solid configurations.

The above discussion emphasizes in an explicit way the deeply different nature of the spin Goldstone mode in the presence of Raman coupling as compared to the spin mode in mixtures with vanishing Raman coupling. In the latter case the identity (47) holds, and employing it in Eq. (48b) one finds

$$\Psi_{s,0,\mathbf{k}}^{(0)}(\mathbf{r},t) \underset{k\to 0}{\sim} e^{-i\mu t/\hbar}(1+i\delta\chi\,\sigma_z)\Psi_0^{(0)}(\mathbf{r}),\tag{50}$$

meaning that the spin mode at $\Omega_R = 0$ represents a mere shift by $\delta\chi = k_R\delta x$ of the relative phase between the two spin components of the wave function. It rotates the direction of the spin polarization vector (see Sec. 3.1) by $\delta\chi$ but has no effect on the (uniform) density of the system.

Result (48b) is particularly remarkable because it shows that in the long-wavelength limit the spin collective mode corresponds to the translational motion of stripes, reflecting the crucial interplay between the spin and density degrees of freedom caused by the spin-orbit coupling. This effect has been recently pointed out numerically in a harmonically trapped spin-orbit-coupled Bose gas by observing that the translational motion of the stripes can be actually induced by the sudden release of a uniform spin perturbation [45].

## 4.4 Sum rules at long wavelengths

Further information about our system can be obtained combining our perturbation approach with the sum-rule method (see Sec. 4.1). We focus on external perturbations proportional to the $\mathbf{q}$-component of either the density operator, $\rho_\mathbf{q} = e^{i\mathbf{q}\cdot\mathbf{r}}$, or the spin-density operator, $s_{z,\mathbf{q}} = \sigma_z e^{i\mathbf{q}\cdot\mathbf{r}}$. The corresponding moments can be deduced from Eq. (36) with $\mathcal{O} = \rho_\mathbf{q}, s_{z,\mathbf{q}}$. Besides the Bogoliubov spectrum $\omega_{b,\mathbf{k}}$, this requires the knowledge of the strengths (35); the perturbative expressions of the latter can be straightforwardly obtained from those of the ground-state wave function $\Psi_0$ and the Bogoliubov amplitudes $\mathcal{W}_{b,\mathbf{k}}$ (see Appendices A and B). It is possible to evaluate any moment at arbitrary $\mathbf{q}$, but here for simplicity we restrict ourselves to the small-$q$ limit. As shown in Eq. (37), the static response is directly related to the inverse-energy-weighted moment. We recall that the $q\to 0$ value of the static density response

coincides with the compressibility, whose second-order expansion takes a simple expression,

$$
\begin{aligned}
\kappa &= \lim_{q \to 0} \chi(\rho_{\mathbf{q}}) \\
&= \frac{1}{G_{dd}} - \frac{4E_R^3}{G_{dd}(2E_R + G_{dd})^3} \left(\frac{\hbar\Omega_R}{4E_R}\right)^2 .
\end{aligned}
\tag{51}
$$

Notice that the relation

$$
\kappa^{-1} = mc_{d,\perp}^2
\tag{52}
$$

holds, reflecting the fact that the phonon mode propagating perpendicular to $x$ exhausts both the inverse-energy-weighted sum rule and the $f$-sum rule [44]

$$
m_1(\rho_{\mathbf{q}}) = N\frac{\hbar^2 q^2}{2m} .
\tag{53}
$$

Instead, the phonon mode along $x$ exhausts the inverse-energy-weighted sum rule but not the $f$-sum rule. In particular we have checked that the ratio between the part of the $f$-sum rule accounted for by the phonon mode as $q \to 0$ and the compressibility sum rule (51) gives direct access to the square of the velocity (44a) of sound waves propagating along the $x$ direction.

The static spin response $\chi(s_{z,\mathbf{q}})$ approaches the magnetic susceptibility as $q \to 0$. At second order in $\hbar\Omega_R/4E_R$ one has

$$
\begin{aligned}
\chi_M &= \lim_{q \to 0} \chi(s_{z,\mathbf{q}}) \\
&= \frac{1}{G_{ss}} + \frac{2G_{dd}E_R^2(2E_R + G_{dd} + G_{ss})}{G_{ss}^2(2E_R + G_{dd})^2(2E_R + G_{ss})} \left(\frac{\hbar\Omega_R}{4E_R}\right)^2 .
\end{aligned}
\tag{54}
$$

The fact that there is no counterpart to Eq. (52) in the spin channel, i.e., $\chi_M^{-1} \neq mc_{s,\perp}^2$, has a simple explanation in terms of sum rules. Indeed, the energy-weighted spin sum rule

$$
m_1(s_{z,\mathbf{q}}) = N\frac{\hbar^2 q^2}{2m} - \hbar\Omega_R\langle\sigma_x\rangle ,
\tag{55}
$$

with $\langle\sigma_x\rangle$ the spin polarization along $x$, is gapped at $q \to 0$ because of the Raman coupling. Thus, it cannot be exhausted by the phonon mode even in the directions perpendicular to $x$.

We plot the inverse compressibility and magnetic susceptibility as functions of $\Omega_R$ in the upper and lower panel of Fig. 3, respectively. As for the results of the previous sections, also here the perturbative formulas match the exact numerical estimates up to relatively large values of the Raman coupling. Notice, in particular, that $\chi_M^{-1}$ vanishes at the spinodal point, pointing out the divergent behavior of the magnetic susceptibility and the occurrence of the dynamic instability discussed in Sec. 4.3. In the bottom panel we additionally compare the obtained magnetic susceptibility with the estimate

$$
\chi_M = \frac{2[16E_R^2 - (\hbar\Omega_R)^2]}{32G_{ss}E_R^2 - (G_{dd} + 2G_{ss})(\hbar\Omega_R)^2} .
\tag{56}
$$

This relation was derived in [61] using a two-harmonic Ansatz for the ground-state wave function, which is expected to be accurate in the limit of small densities. Notice that Eq. (56) diverges at the critical Raman coupling (7), reflecting the fact that the transition and the spinodal point tend to coincide at low densities. In addition, when the two conditions $G_{dd,ss} \ll E_R$ and $\hbar\Omega_R \ll 4E_R$ are simultaneously satisfied, Eqs. (54) and (56) reduce to the same expression.

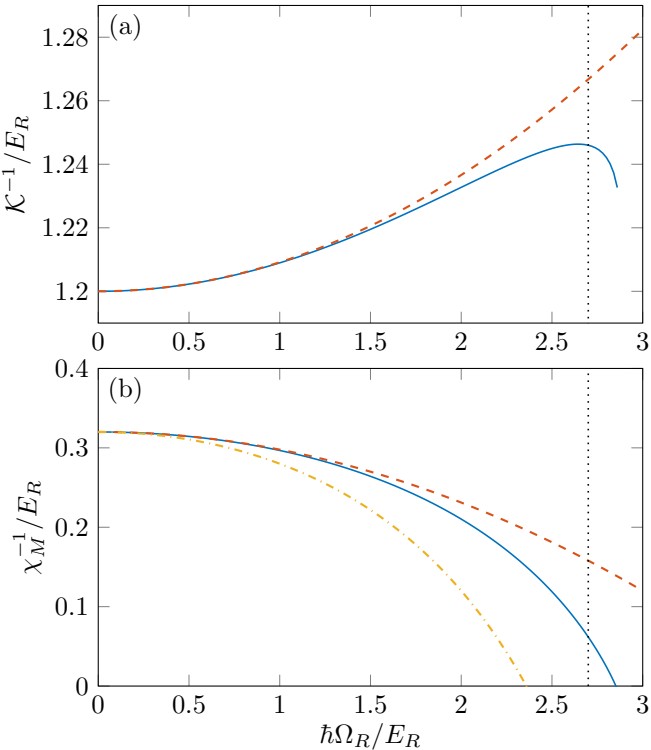

Figure 3: (a) inverse compressibility and (b) inverse magnetic susceptibility as functions of the Raman coupling $\Omega_R$. The meaning of the lines is the same as in Fig. 1: the blue solid ones show the exact values determined numerically, while the red dashed ones correspond to the perturbation results up to second order in $\hbar\Omega_R/4E_R$. The yellow dash-dot line in (b) corresponds to the low-density prediction (56) and vanishes at $\hbar\Omega_R = 2.35\,E_R$. The vertical black dotted lines at $\hbar\Omega_R = 2.70\,E_R$ identify the transition to the plane-wave phase. The interaction parameters are the same as Fig. 1.

## 4.5 Superfluid density

The results derived in the previous sections concerning the sound velocities and the sum rules can be usefully employed to calculate the superfluid density of a spin-orbit-coupled Bose gas in the stripe (supersolid) phase. We will follow the procedure developed in [62], based on Baym's approach [63] to the normal (non superfluid) density in terms of the macroscopic limit of the static response to the transverse current field:

$$
\begin{aligned}
\frac{\rho_n}{\bar{\rho}} &= m \lim_{q\to 0} \chi(J_{\mathbf{q},x}^{\perp}) \\
&= \frac{m}{N} \lim_{q\to 0} \sum_{b,\mathbf{k}} \left(\hbar\omega_{b,\mathbf{k}}\right)^{-1} \left( |\langle 0|J_{\mathbf{q},x}^{\perp}|b,\mathbf{k}\rangle|^2 + |\langle 0|J_{-\mathbf{q},x}^{\perp}|b,\mathbf{k}\rangle|^2 \right),
\end{aligned}
\tag{57}
$$

where $\bar{\rho} = m\bar{n}$ is the average mass density and

$$
J_{\mathbf{q},x}^{\perp} = \frac{1}{m}(p_x - \hbar k_R \sigma_z)e^{i\mathbf{q}_{\perp}\cdot\mathbf{r}}
\tag{58}
$$

is the transverse current operator along the $x$ direction (here $\mathbf{q}_{\perp}$ is taken perpendicular to $x$). A crucial feature of spin-orbit-coupled gases is that the current along the $x$ direction contains a novel spin-dependent term which is responsible for the violation of Galilean invariance.

For this reason, even for uniform-density configurations, like the plane-wave and the zero-momentum phases, the superfluid density $\rho_s = \bar{\rho} - \rho_n$, calculated at zero temperature, does not coincide with the total density [62] and is expected to be an anisotropic tensor since the $y$ and $z$ components of the current are not affected by the spin term.

Since the transverse current operator does not excite the gapless phonon mode, which is of longitudinal nature, the only contribution to Eq. (57) comes from the gapped branch. On the other hand, the $q \to 0$ contribution associated with the gapped modes can be safely calculated by replacing $\mathbf{q}_\perp$ with $q_x \mathbf{e}_x$, a procedure that would not be allowed in the calculation of the $q \to 0$ contributions of the gapless excitations. Making explicit use of the equation of continuity to relate the static longitudinal current response to the energy-weighted $f$-sum rule (53), one then finds that the normal density is directly related to the contribution provided by the gapped excitations to the $f$-sum rule. By taking the difference $\rho_s = \bar{\rho} - \rho_n$ and noticing that the phonon modes exhaust the inverse-energy-weighted moment, fixed by the compressibility $\kappa$ of the system (see Sec. 4.4), one finds the following useful relationship between the superfluid density, the velocity of sound, and the compressibility holding at zero temperature:

$$\frac{\rho_s^x}{\bar{\rho}} = mc_{d,x}^2 \kappa. \tag{59}$$

Analogously one finds the result $\rho_s^y/\bar{\rho} = \rho_s^z/\bar{\rho} = mc_{d,\perp}^2 \kappa$ for the superfluid density calculated for the motion parallel to the stripes ($y$ and $z$ directions). It reveals, as expected, that the superfluid density is not isotropic. As shown in Sec. 4.3, the sound velocities along the $x$ and $y$ (or $z$) directions are actually different. Since, as already pointed out in the previous section, the phonon mode propagating along the directions perpendicular to $x$ exhausts the corresponding $f$-sum rule, the equation $\kappa^{-1} = mc_{d,\perp}^2$ holds and one then concludes that the value of the superfluid density can be determined from the ratio of the sound velocities propagating parallel and perpendicular to the $x$ direction, according to

$$\frac{\rho_s^x}{\bar{\rho}} = \frac{c_{d,x}^2}{c_{d,\perp}^2}. \tag{60}$$

Result (60) holds independently of the value of the Raman coupling $\Omega_R$, in the stripe as well as in the other quantum phases exhibited by spin-orbit-coupled BECs, provided that the symmetric intraspecies coupling case ($g_{\uparrow\uparrow} = g_{\downarrow\downarrow}$) is considered. In Fig. 4 we report the value of the ratio (60) calculated either employing the $(\hbar\Omega_R/4E_R)^2$ expansion using the results (44a)-(45a) for the sound velocities, and the full results employing the exact values for $c_{d,x}^2$ and $c_{d,\perp}^2$ reported in Fig. 2. Result (60) is also well suited to determine experimentally the superfluid density of spin-orbit-coupled Bose gases through the direct measurements of the sound velocities. In the figure we also show the results for the superfluid density calculated in the plane-wave and zero-momentum phases, using the findings of [49] for the sound velocities and clearly revealing the first-order nature of the phase transition between the stripe and the plane-wave phase. Notice that in the plane-wave phase the sound velocity $c_{d,x}^2$ should be actually replaced by the product $c_{x,+}c_{x,-}$ of the sound velocities along the $+x$ and $-x$ directions. The general behavior of the superfluid density as a function of the Raman coupling $\Omega_R$ is qualitatively similar to the results derived in [51], where the phase twist method was employed using a variational Ansatz for the order parameter in the stripe phase; first studies based on Monte Carlo methods have shown that quantum fluctuations do not significantly affect the value of the superfluid density, which in three-dimensional dilute systems remains close to the mean-field prediction [54]. As pointed out in [62] the effects of the spin-orbit coupling have a dramatic consequence near the second-order transition between the plane-wave and zero-momentum phases, characterized by the divergence of the magnetic susceptibility $\chi_M$. In the

same paper the theoretical predictions for the superfluid density were successfully compared with experiments in both the plane-wave and zero-momentum phases. The present approach generalizes the study of the superfluid density to the supersolid stripe phase, suggesting a direct procedure for its experimental measurement.

We finally compare the above results with those obtained from the Leggett criterion [64, 65]. According to this criterion, the superfluid fraction of a system exhibiting density modulations along $x$, with periodicity $\pi/k_1$, is bounded from above by the quantity

$$\frac{\rho^x_{s,L}}{\bar{\rho}} = \left[ \frac{1}{\pi/k_1} \int_{-\pi/2k_1}^{\pi/2k_1} \frac{dx}{\rho(x)/\bar{\rho}} \right]^{-1} , \tag{61}$$

with $\rho = mn$ the local mass density (here assumed to be a function of the sole variable $x$). Leggett deduced this upper bound using a class of Ansatz many-body wave functions where all the particles in the superfluid have the same phase. The mean-field Ansatz belongs to this class, as it makes the stronger assumption that all the particles have identical wave functions. Consequently, within the mean-field approximation the upper bound $\rho^x_{s,L}$ is always saturated, i.e., it coincides with the real superfluid density $\rho^x_s$. However, our system represents an exception to this rule because the single-particle Hamiltonian (1) lacks time-reversal invariance, which is another requirement for the derivation of Leggett's criterion. Hence, Eqs. (60) and (61) can give different results in the case of spin-orbit-coupled Bose gases. This is obvious in the uniform phases, where Leggett's criterion predicts that the superfluid density coincides with the total density, $\rho^x_{s,L}/\bar{\rho} = 1$, in stark contrast with the correct value reported in [62] and shown in Fig. 4. A similar effect takes place in the stripe phase, where $\rho^x_{s,L}/\bar{\rho}$ is not one but still significantly larger than the correct superfluid density $\rho^x_s/\bar{\rho}$, see the blue dash-dot line in Fig. 4. This line actually corresponds to the exact numerical evaluation of the integral (61), but one can also compute it within the perturbative framework of the present work. The final results up to second order in $(\hbar\Omega_R/4E_R)^2$ can be cast in the form

$$\frac{\rho^x_{s,L}}{\bar{\rho}} = 1 - \frac{\mathcal{C}^2}{2} , \tag{62}$$

where the contrast $\mathcal{C}$ is given by Eq. (18). This simple relation between Leggett's upper bound and the contrast of fringes is a common feature of shallow one-dimensional supersolids; it was deduced by general arguments in Ref. [66], and directly proven in the case of cnoidal waves [41]. Our perturbative approach enabled us to check its validity in spin-orbit-coupled Bose gases.

# 5 Conclusion

We carried out a detailed analysis of a spin-orbit-coupled Bose-Einstein condensate in the stripe phase. By developing a suitable perturbation approach we obtained analytical expressions for a number of observables of interest. In particular, for systems at equilibrium we computed the periodicity and contrast of the stripes, as well as the chemical potential and energy per particle. Although our formulas are derived in the regime of small Raman coupling, they turn out to be quantitatively accurate also for significantly large values of the coupling.

The extension of the perturbation method to the Bogoliubov modes enabled us to compute with similar accuracy the dispersion relation of the elementary excitations. We focused on the long-wavelength limit, where we proved that the two lowest-lying bands are gapless despite the presence of the Raman coupling term. We evaluated the velocities of sound waves propagating perpendicular and parallel to the stripes, as well as the corresponding Bogoliubov

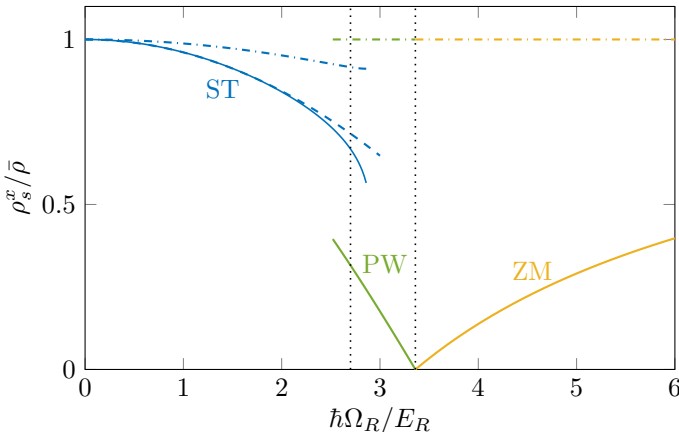

Figure 4: Superfluid density estimated from Eq. (60) as a function of the Raman coupling $\Omega_R$. The solid blue, green, and yellow lines show the exact values in the stripe (ST), plane-wave (PW), and zero-momentum (ZM) phase, respectively; the blue dashed line corresponds to the perturbation results in the stripe phase up to second order in $\hbar\Omega_R/4E_R$. For each phase we also report the estimates obtained from the Leggett criterion (61) (dash-dot curves). The vertical black dotted lines at $\hbar\Omega_R = 2.70\, E_R$ and $\hbar\Omega_R = 3.36\, E_R$ identify the transitions between the various phases. Besides the spinodal behavior of the stripe phase we previously pointed out, here we see that the plane-wave phase can be found as a metastable state down to $\hbar\Omega_R = 2.52\, E_R$, where the roton gap in its excitation spectrum vanishes (see Ref. [49]). The interaction parameters are the same as Fig. 1.

amplitudes, that reveal the Goldstone nature of these modes. In particular, while in the long-wavelength limit the density mode is dominated by a mere rotation of the global phase of the order parameter, the spin mode corresponds to a translation of the density fringes with respect to the equilibrium position. The simultaneous presence of a superfluid and a crystal mode is a typical signature of supersolidity. It also marks a striking difference with respect to binary mixtures without spin-orbit coupling, where the spin mode represents a simple rotation of the spin polarization.

We subsequently calculated the most relevant sum rules in the long-wavelength regime, yielding the compressibility, the magnetic susceptibility, and the superfluid density. The latter was directly related to the ratio of the sound velocities along and perpendicular to the direction of the spin-orbit coupling. This provides a viable way to measure the superfluid density of spin-orbit-coupled Bose-Einstein condensates using currently available experimental setups. Comparing the results of this method to those of the Leggett criterion we found a significant discrepancy, that highlights the lack of time-reversal invariance in our system.

# Acknowledgements

Many useful discussions and collaborations with Yun Li, Kevin T. Geier, and Philipp Hauke are acknowledged.

**Funding information** The research leading to these results has received funding from the European Research Council under European Community's Seventh Framework Programme (FP7/2007-2013 Grant Agreement No. 341197).

# A  Perturbation calculation of the ground-state wave function

In this appendix we delineate the general procedure for determining the ground-state wave function at arbitrary order in the Raman coupling (Sec. A.1). Subsequently, we explicitly compute the first- (Sec. A.2) and second-order (Sec. A.3) corrections.

## A.1  General procedure and recurrence relations for the ground state

As mentioned in Sec. 3.2, after inserting the expansions (15), (16), and (17) into the Gross-Pitaevskii equation (12) and equating terms of the same order on both sides, one deduces for $n \geq 1$ the recurrence relation

$$\left[E_R(-i\nabla_x - \sigma_z)^2 + \mathcal{L}_D\right]\Psi_0^{(n)} + \mathcal{L}_C\Psi_0^{(n)*} = \mu^{(n)}\Psi_0^{(0)} - \mathcal{J}^{(n)}. \tag{63}$$

On the left-hand side of this equation one has the two matrices

$$\mathcal{L}_D = \frac{G_{dd} + G_{ss}}{2}\mathbb{I}_2 + \frac{G_{dd} - G_{ss}}{2}(e^{2ix}\sigma_+ + e^{-2ix}\sigma_-), \tag{64a}$$

$$\mathcal{L}_C = \frac{G_{dd} - G_{ss}}{2}\sigma_x + \frac{G_{dd} + G_{ss}}{2}(e^{2ix}\sigma_\uparrow + e^{-2ix}\sigma_\downarrow), \tag{64b}$$

where $\sigma_\pm = (\sigma_x \pm i\sigma_y)/2$, $\sigma_{\uparrow,\downarrow} = (\mathbb{I}_2 \pm \sigma_z)/2$, and $\mathbb{I}_2$ is the $2\times 2$ identity matrix. For simplicity, in the calculations of this Appendix and all the next ones we take $\chi_0 = 0$ in the unperturbed wave function (14). On the right-hand side of Eq. (63) one has a source term $\mathcal{J}^{(n)}$ that only depends on quantities determined up to order $n-1$. It reads

$$
\begin{aligned}
\mathcal{J}^{(n)} = & -\frac{\hbar^2}{2m}\sum_{l'=2}^{n}\sum_{l=1}^{l'-1}k_1^{(l)}k_1^{(l'-l)}\nabla_x^2\Psi_0^{(n-l')} + \frac{\hbar^2 k_R}{m}\sum_{l=1}^{n-1}k_1^{(l)}\left(-\nabla_x^2 + i\sigma_z\nabla_x\right)\Psi_0^{(n-l)} \\
& + \frac{\hbar\Omega_R}{2}\sigma_x\Psi_0^{(n-1)} - \sum_{l=1}^{n-1}\mu^{(l)}\Psi_0^{(n-l)} + g_{dd}\sum_{l,l',l''=0}^{n-1}\left[\Psi_0^{(l)\dagger}\Psi_0^{(l')}\right]\Psi_0^{(l'')}\delta_{n,l+l'+l''} \\
& + g_{ss}\sum_{l,l',l''=0}^{n-1}\left[\Psi_0^{(l)\dagger}\sigma_z\Psi_0^{(l')}\right]\sigma_z\Psi_0^{(l'')}\delta_{n,l+l'+l''}.
\end{aligned}
\tag{65}
$$

In the next two sections we shall consider the cases $n = 1$ and $n = 2$, where this involved formula reduces to much simpler expressions. Notice that, before solving Eq. (63) for a given $n$, one has first to determine $\mu^{(n)}$. This can be done by multiplying both sides of Eq. (63) by $\Psi_0^{(0)\dagger}$ and integrating with respect to the spatial coordinates. The resulting expression will contain terms depending on $\Psi_0^{(n)}$, that can be eliminated using the relation

$$\int_V d^3r\left[\Psi_0^{(0)\dagger}\Psi_0^{(n)} + \Psi_0^{(n)\dagger}\Psi_0^{(0)}\right] = -\int_V d^3r\sum_{l=1}^{n-1}\Psi_0^{(l)\dagger}\Psi_0^{(n-l)}, \tag{66}$$

that follows from the normalization condition (5).

Since Eq. (63) is linear, any of its solutions can be expressed as the sum of a particular solution and of the general integral of the associated homogeneous equation. This general integral reads

$$
\begin{aligned}
\Psi_{GI}^{(n)} = & \left[A_{d,+}^{(n)}e^{\sqrt{2G_{dd}/E_R}x} + A_{d,-}^{(n)}e^{-\sqrt{2G_{dd}/E_R}x}\right]\Psi_0^{(0)} \\
& + \left[A_{s,+}^{(n)}e^{\sqrt{2G_{ss}/E_R}x} + A_{s,-}^{(n)}e^{-\sqrt{2G_{ss}/E_R}x}\right]\sigma_z\Psi_0^{(0)} \\
& + i\left[B_{d,1}^{(n)} + B_{s,1}^{(n)}\sigma_z\right]x\,\Psi_0^{(0)} \\
& + i\left[B_{d,0}^{(n)} + B_{s,0}^{(n)}\sigma_z\right]\Psi_0^{(0)}.
\end{aligned}
\tag{67}
$$

Here the $A$'s and $B$'s constitute a set of 8 real arbitrary constants. In order to fix their values we first require that the wave function of the stripe phase be $2\pi$-periodic in $x$ at any order in $\hbar\Omega_R/4E_R$. This implies that the coefficients of the nonperiodic terms in Eq. (67) have to vanish, i.e., $A_{d,\pm}^{(n)} = A_{s,\pm}^{(n)} = B_{d,1}^{(n)} = B_{s,1}^{(n)} = 0$. Regarding the periodic part of the general integral (67), corresponding to its last row, one can easily verify that the first term merely changes the global phase of the wave function, while the second term produces a shift of the offset of the density fringes. We recall that, because of the U(1) and translation symmetry of the model, if $\Psi_0(x,y,z)$ is a solution of Eq. (12), so is $e^{i\varphi_0}\Psi_0(x-x_0,y,z)$ for arbitrary $\varphi_0$ and $x_0$. We choose the values of these two parameters by requiring the additional constraints

$$\int_V d^3r \, \text{Im}\left[\Psi_0^{(0)\dagger}\Psi_0^{(n)}\right] = 0, \tag{68a}$$

$$\int_V d^3r \, \text{Im}\left[\Psi_0^{(0)\dagger}\sigma_z\Psi_0^{(n)}\right] = 0 \tag{68b}$$

to be fulfilled at all perturbation orders. This corresponds to taking $B_{d,0}^{(n)} = B_{s,0}^{(n)} = 0$ in Eq. (67).

## A.2 First-order corrections to the ground state

Using the prescriptions of the previous section one can easily show that the first-order correction to the chemical potential is vanishing, $\mu^{(1)} = 0$. Besides, the source term (65) with $n = 1$ is simply

$$\mathcal{J}^{(1)} = \frac{\hbar\Omega_R}{2}\sigma_x\Psi_0^{(0)} = \sqrt{\frac{\bar{n}}{2}}\left[\begin{pmatrix} 0 \\ \hbar\Omega_R/2 \end{pmatrix}e^{ix} + \begin{pmatrix} \hbar\Omega_R/2 \\ 0 \end{pmatrix}e^{-ix}\right]. \tag{69}$$

The solution of Eq. (63) with $n = 1$ that satisfies the conditions (68) has the form

$$\Psi_0^{(1)} = \sqrt{\bar{n}}\left[\begin{pmatrix} \tilde{\Psi}_{+3,\uparrow}^{(1)} \\ 0 \end{pmatrix}e^{3ix} + \begin{pmatrix} 0 \\ \tilde{\Psi}_{+1,\downarrow}^{(1)} \end{pmatrix}e^{ix} + \begin{pmatrix} \tilde{\Psi}_{-1,\uparrow}^{(1)} \\ 0 \end{pmatrix}e^{-ix} + \begin{pmatrix} 0 \\ \tilde{\Psi}_{-3,\downarrow}^{(1)} \end{pmatrix}e^{-3ix}\right]. \tag{70}$$

Here the coefficients can be determined by inserting Eq. (70) into (63) and equating the terms on both sides with the same oscillating behavior. This yields

$$\tilde{\Psi}_{+3,\uparrow}^{(1)} = \tilde{\Psi}_{-3,\downarrow}^{(1)} = \frac{G_{dd}}{4\sqrt{2}(2E_R + G_{dd})}\frac{\hbar\Omega_R}{4E_R}, \tag{71a}$$

$$\tilde{\Psi}_{+1,\downarrow}^{(1)} = \tilde{\Psi}_{-1,\uparrow}^{(1)} = -\frac{4E_R + G_{dd}}{4\sqrt{2}(2E_R + G_{dd})}\frac{\hbar\Omega_R}{4E_R}. \tag{71b}$$

## A.3 Second-order corrections to the ground state

With the results of Sec. A.2 at hand, the second-order correction to the chemical potential is easily evaluated and found equal to the second term on the right-hand side of Eq. (19) of the main text. The source term (65) with $n = 2$ takes the form

$$\mathcal{J}^{(2)} = \sqrt{\bar{n}}\left[\begin{pmatrix} \mathcal{J}_5^{(2)} \\ 0 \end{pmatrix}e^{5ix} + \begin{pmatrix} 0 \\ \mathcal{J}_3^{(2)} \end{pmatrix}e^{3ix} + \begin{pmatrix} \mathcal{J}_1^{(2)} \\ 0 \end{pmatrix}e^{ix} \right.$$
$$\left. + \begin{pmatrix} 0 \\ \mathcal{J}_1^{(2)} \end{pmatrix}e^{-ix} + \begin{pmatrix} \mathcal{J}_3^{(2)} \\ 0 \end{pmatrix}e^{-3ix} + \begin{pmatrix} 0 \\ \mathcal{J}_5^{(2)} \end{pmatrix}e^{-5ix}\right], \tag{72}$$

where

$$\mathcal{J}_5^{(2)} = -\frac{G_{dd}^2(8E_R + G_{dd})}{\sqrt{2}[4(2E_R + G_{dd})]^2}\left(\frac{\hbar\Omega_R}{4E_R}\right)^2, \tag{73a}$$

$$\mathcal{J}_3^{(2)} = \frac{E_R(32E_R^2 + 8G_{dd}E_R - G_{dd}^2)}{\sqrt{2}[4(2E_R + G_{dd})]^2}\left(\frac{\hbar\Omega_R}{4E_R}\right)^2, \tag{73b}$$

$$\mathcal{J}_1^{(2)} = -\frac{32E_R^3 + 8G_{dd}E_R^2 + G_{dd}^3}{\sqrt{2}[4(2E_R + G_{dd})]^2}\left(\frac{\hbar\Omega_R}{4E_R}\right)^2. \tag{73c}$$

The solution of Eq. (63) with $n = 2$ has the structure

$$\Psi_0^{(2)} = \sqrt{\bar{n}}\Bigg[\begin{pmatrix}\tilde{\Psi}_{+5,\uparrow}^{(2)}\\0\end{pmatrix}e^{5ix} + \begin{pmatrix}0\\\tilde{\Psi}_{+3,\downarrow}^{(2)}\end{pmatrix}e^{3ix} + \begin{pmatrix}\tilde{\Psi}_{+1,\uparrow}^{(2)}\\0\end{pmatrix}e^{ix}$$
$$+ \begin{pmatrix}0\\\tilde{\Psi}_{-1,\downarrow}^{(2)}\end{pmatrix}e^{-ix} + \begin{pmatrix}\tilde{\Psi}_{-3,\uparrow}^{(2)}\\0\end{pmatrix}e^{-3ix} + \begin{pmatrix}0\\\tilde{\Psi}_{-5,\downarrow}^{(2)}\end{pmatrix}e^{-5ix}\Bigg]. \tag{74}$$

The procedure for determining the coefficients is the same as the previous section, and yields

$$\tilde{\Psi}_{+5,\uparrow}^{(2)} = \tilde{\Psi}_{-5,\downarrow}^{(2)} = \frac{G_{dd}^2(5E_R + G_{dd})}{\sqrt{2}(8E_R + G_{dd})[4(2E_R + G_{dd})]^2}\left(\frac{\hbar\Omega_R}{4E_R}\right)^2, \tag{75a}$$

$$\tilde{\Psi}_{+3,\downarrow}^{(2)} = \tilde{\Psi}_{-3,\uparrow}^{(2)} = -\frac{G_{dd}E_R(16E_R + 5G_{dd})}{\sqrt{2}(8E_R + G_{dd})[4(2E_R + G_{dd})]^2}\left(\frac{\hbar\Omega_R}{4E_R}\right)^2, \tag{75b}$$

$$\tilde{\Psi}_{+1,\uparrow}^{(2)} = \tilde{\Psi}_{-1,\downarrow}^{(2)} = -\frac{8E_R^2 + 4G_{dd}E_R + G_{dd}^2}{\sqrt{2}[4(2E_R + G_{dd})]^2}\left(\frac{\hbar\Omega_R}{4E_R}\right)^2. \tag{75c}$$

These results enable us to determine the second-order correction to $k_1$ [from Eq. (11)], and to the energy per particle [from Eq. (6)], which are provided in Eq. (21) and (20) of the main text, respectively.

We point out that the perturbative corrections given in Eqs. (70) and (74) feature harmonic components that do not correspond to minima of the single-particle dispersion, their appearance being uniquely caused by the interaction. In general, the $n$-th–order correction to the condensate wave function contains harmonic terms with wave vectors ranging from $\pm 1$ to $\pm(2n + 1)$ (in units of $k_1$). It is natural to expect that the perturbation series for $\Psi_0$ converges to the Bloch-wave structure (9). Although evidence for the presence of such higher-order harmonics was first found numerically in Ref. [36], the perturbation approach employed in the present work has the advantage of providing a better insight into their origin.

# B  Perturbation calculation of the Bogoliubov modes

The purpose of this appendix is to present the perturbation scheme for studying the Bogoliubov modes in the stripe phase. We start by introducing the band structure of the spectrum at vanishing Raman coupling (Sec. B.1) and deriving the recurrence relations (Sec. B.2). Then, we compute the perturbative corrections to the Bogoliubov frequencies and amplitudes at first (Sec. B.3) and second (Sec. B.4) order in the Raman coupling.

## B.1  Band structure at $\Omega_R = 0$

As we mentioned at the end of Sec. 4.2, the Bogoliubov modes at $\Omega_R = 0$ can be labeled either by the physical momentum $\mathbf{q}$ or the quasimomentum $\mathbf{k}$. Although the latter possibility may

look unnatural, it is an obvious choice in view of the perturbation study of the Bogoliubov solutions at finite Raman coupling. One can define $\mathbf{k}$ at vanishing $\Omega_R$ by rewriting the components of $\mathbf{q}$ as

$$q_x = k_x + \bar{K}, \quad q_y = k_y, \quad q_z = k_z. \tag{76}$$

Note that the $x$ component of the quasimomentum, $k_x = 2\{q_x/2\}$, coincides with $q_x$ up to a reciprocal lattice vector, $\bar{K} = 2[q_x/2]$. Here we have used the standard notation $\{\cdots\}$ and $[\cdots]$ for the fractional and integer part of a real number, respectively. On the one hand, these definitions ensure that $k_x$ is confined in the first Brillouin zone, i.e., $0 \le k_x < 2$ for any $q_x$; on the other hand, one has that $\bar{K}$ is always an integer multiple of the size of the Brillouin zone, equal to 2 when expressed in units of $k_1$.

Let us insert Eq. (76) into Eqs. (38), (39), and (40), and label all quantities by $\bar{K}$ and $\mathbf{k}$ instead of $\mathbf{q}$. In this new representation the Bogoliubov spectrum (38) exhibits an infinite number of branches, each labeled by two indices $\ell = d,s$ and $\bar{K}$, and restricted to the first Brillouin zone. In Fig. 5 we plot some of the lowest-lying branches as functions of $k_x$ (we take $k_y = k_z = 0$). Notice that a band structure with two gapless modes is already visible in this figure, although there are no gaps separating the various bands because $\Omega_R$ is zero.

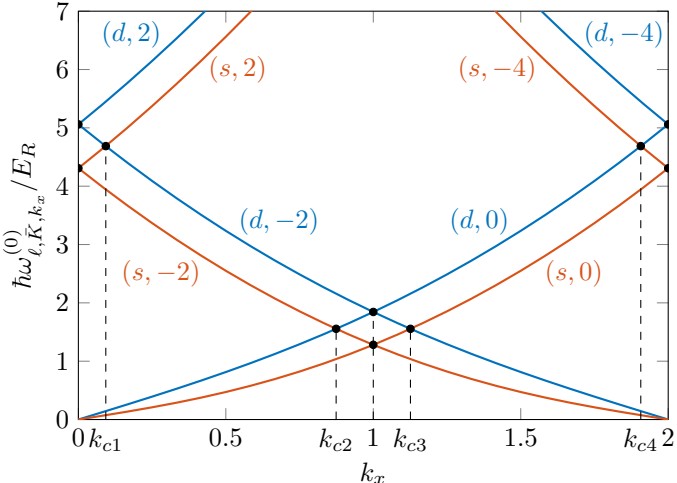

Figure 5: Lowest bands of the Bogoliubov spectrum as functions of $k_x$ at $\Omega_R = 0$. Blue and red curves correspond to density and spin modes, respectively; for better clarity, close to each band we write the corresponding values of the indices $(\ell, \bar{K})$. The black dots mark the band crossing points, where a gap is expected to open as soon as $\Omega_R \ne 0$. The interaction parameters are the same as Fig. 1.

As a final step we replace the couple of indices $(\ell, \bar{K})$ with the single band index $b$ introduced at the end of Sec. 4.1. This can be accomplished by sorting all the $\omega_{\ell,\bar{K},\mathbf{k}}^{(0)}$'s in ascending order at fixed $\mathbf{k}$; we denote the ordered Bogoliubov bands by $\omega_{b,\mathbf{k}}^{(0)}$, and the corresponding amplitudes by $\mathcal{W}_{b,\mathbf{k}}^{(0)}$. In performing the above procedure one has to take into account that the $\omega_{\ell,\bar{K},\mathbf{k}}^{(0)}$'s can cross one another (see Fig. 5), meaning that how to arrange them depends on $\mathbf{k}$. For instance, for the two lowest bands shown in Fig. 5 one has

$$\omega_{b=1,k_x}^{(0)} = \begin{cases} \omega_{s,0,k_x}^{(0)} & \text{if } 0 \le k_x \le 1 \\ \omega_{s,-2,k_x}^{(0)} & \text{if } 1 \le k_x \le 2 \end{cases}, \quad \omega_{b=2,k_x}^{(0)} = \begin{cases} \omega_{d,0,k_x}^{(0)} & \text{if } 0 \le k_x \le k_{c2} \\ \omega_{s,-2,k_x}^{(0)} & \text{if } k_{c2} \le k_x \le 1 \\ \omega_{s,0,k_x}^{(0)} & \text{if } 1 \le k_x \le k_{c3} \\ \omega_{d,-2,k_x}^{(0)} & \text{if } k_{c3} \le k_x \le 2 \end{cases}, \tag{77}$$

where $k_{c2}$ and $k_{c3}$ are the crossing points marked in the figure.[2]

We conclude this section by mentioning that the Bogoliubov amplitudes $\mathcal{W}_{b,\mathbf{k}}^{(0)}$ and $\bar{\mathcal{W}}_{b,\mathbf{k}}^{(0)}$ obey the orthonormalization conditions

$$\int_V d^3r\, \mathcal{W}_{b,\mathbf{k}}^{(0)\dagger} \eta \mathcal{W}_{b',\mathbf{k}'}^{(0)} = -\int_V d^3r\, \bar{\mathcal{W}}_{b,\mathbf{k}}^{(0)\dagger} \eta \bar{\mathcal{W}}_{b',\mathbf{k}'}^{(0)} = \delta_{bb'}\delta_{\mathbf{k}\mathbf{k}'}, \tag{78a}$$

$$\int_V d^3r\, \bar{\mathcal{W}}_{b,\mathbf{k}}^{(0)\dagger} \eta \mathcal{W}_{b',\mathbf{k}'}^{(0)} = \int_V d^3r\, \mathcal{W}_{b,\mathbf{k}}^{(0)\dagger} \eta \bar{\mathcal{W}}_{b',\mathbf{k}'}^{(0)} = 0, \tag{78b}$$

as well as the completeness relation

$$\sum_{b,\mathbf{k}} \left[ \mathcal{W}_{b,\mathbf{k}}^{(0)}(\mathbf{r}) \mathcal{W}_{b,\mathbf{k}}^{(0)\dagger}(\mathbf{r}') - \bar{\mathcal{W}}_{b,\mathbf{k}}^{(0)}(\mathbf{r}) \bar{\mathcal{W}}_{b,\mathbf{k}}^{(0)\dagger}(\mathbf{r}') \right] = \eta \delta(\mathbf{r}' - \mathbf{r}). \tag{79}$$

We recall that, because of the symmetry properties of the operator $\mathcal{B}$ [see Eq. (29)], if $\mathcal{W}_{b,\mathbf{k}}^{(0)}$ is solution of Eq. (27) at $\Omega_R = 0$ with frequency $\omega_{b,\mathbf{k}}^{(0)}$, then $\bar{\mathcal{W}}_{b,\mathbf{k}}^{(0)}$ is also solution with frequency $-\omega_{b,\mathbf{k}}^{(0)}$ [59]. From Eq. (78a) we see that the two solutions have opposite norm. However, they actually correspond to the same physical oscillation of the system. Hence, without loss of generality we will restrict our perturbation analysis to the sole positive-norm modes. Nevertheless, the contribution of the negative-norm modes is crucial when writing the completeness relation (79). These properties will play a fundamental role in the formulation of the perturbation approach for the Bogoliubov modes.

## B.2 General procedure and recurrence relations for the Bogoliubov modes

Let us now insert the expansions (42) and (43) into Eq. (27). To order $n$ one obtains

$$\sum_{l=0}^{n} \mathcal{B}^{(l)} \mathcal{W}_{b,\mathbf{k}}^{(n-l)} = \hbar \sum_{l=0}^{n} \omega_{b,\mathbf{k}}^{(l)} \eta \mathcal{W}_{b,\mathbf{k}}^{(n-l)}. \tag{80}$$

This recurrence relation is the starting point for our perturbative calculation of the Bogoliubov spectrum. As we will show in the next sections, an important building block of our formalism is represented by the integrals

$$\mathcal{B}_{b'b,\mathbf{k}'\mathbf{k}}^{(n)} = \int_V d^3r\, \mathcal{W}_{b',\mathbf{k}'}^{(0)\dagger} \mathcal{B}^{(n)} \mathcal{W}_{b,\mathbf{k}}^{(0)}, \tag{81a}$$

$$\mathcal{B}_{\overline{b'}b,\overline{\mathbf{k}'}\mathbf{k}}^{(n)} = \int_V d^3r\, \bar{\mathcal{W}}_{b',\mathbf{k}'}^{(0)\dagger} \mathcal{B}^{(n)} \mathcal{W}_{b,\mathbf{k}}^{(0)}, \tag{81b}$$

and analogous expressions for $\mathcal{B}_{b'\overline{b},\mathbf{k}'\overline{\mathbf{k}}}^{(n)}$ and $\mathcal{B}_{\overline{b'}\overline{b},\overline{\mathbf{k}'}\overline{\mathbf{k}}}^{(n)}$. These quantities can be calculated by expressing $\mathcal{W}_{b,\mathbf{k}}^{(0)}$ and $\bar{\mathcal{W}}_{b,\mathbf{k}}^{(0)}$ in terms of $\mathcal{W}_{\ell,\bar{K},\mathbf{k}}$ and $\bar{\mathcal{W}}_{\ell,\bar{K},\mathbf{k}}$ according to the above prescriptions. One finds that only a limited subset of these integrals are nonzero; some of those that are needed for the calculations of the present work are reported in Appendix C. Notice that they vanish when $\mathbf{k}' \neq \mathbf{k}$ [for $\mathcal{B}_{b'b,\mathbf{k}'\mathbf{k}}^{(n)}$ and $\mathcal{B}_{\overline{b'}\overline{b},\overline{\mathbf{k}'}\overline{\mathbf{k}}}^{(n)}$] or $\mathbf{k}' \neq -\mathbf{k}$ [for $\mathcal{B}_{\overline{b'}b,\overline{\mathbf{k}'}\mathbf{k}}^{(n)}$ and $\mathcal{B}_{b'\overline{b},\mathbf{k}'\overline{\mathbf{k}}}^{(n)}$]. The knowledge of the integrals (81) is crucial to compute the perturbative corrections to the Bogoliubov

---

[2]Notice that Eq. (77) is valid only if $G_{dd} > G_{ss}$.

frequencies and amplitudes. For the latter we will use the completeness relation (79) to express $\mathcal{W}_{b,\mathbf{k}}^{(n)}$ as a linear combination of the $\mathcal{W}_{b,\mathbf{k}}^{(0)}$'s and the $\bar{\mathcal{W}}_{b,\mathbf{k}}^{(0)}$'s,

$$
\begin{aligned}
\mathcal{W}_{b,\mathbf{k}}^{(n)}(\mathbf{r}) = \sum_{b',\mathbf{k}'} & \left[ \int_V d^3r' \, \mathcal{W}_{b',\mathbf{k}'}^{(0)\dagger}(\mathbf{r}')\eta \mathcal{W}_{b,\mathbf{k}}^{(n)}(\mathbf{r}') \right] \mathcal{W}_{b',\mathbf{k}'}^{(0)}(\mathbf{r}) \\
& - \sum_{b',\mathbf{k}'} \left[ \int_V d^3r' \, \bar{\mathcal{W}}_{b',\mathbf{k}'}^{(0)\dagger}(\mathbf{r}')\eta \mathcal{W}_{b,\mathbf{k}}^{(n)}(\mathbf{r}') \right] \bar{\mathcal{W}}_{b',\mathbf{k}'}^{(0)}(\mathbf{r}) .
\end{aligned}
\tag{82}
$$

### B.3 First-order corrections to the Bogoliubov modes

Let us consider Eq. (80) with $n = 1$,

$$
\mathcal{B}^{(0)}\mathcal{W}_{b,\mathbf{k}}^{(1)} + \mathcal{B}^{(1)}\mathcal{W}_{b,\mathbf{k}}^{(0)} = \hbar\omega_{b,\mathbf{k}}^{(0)}\eta \mathcal{W}_{b,\mathbf{k}}^{(1)} + \hbar\omega_{b,\mathbf{k}}^{(1)}\eta \mathcal{W}_{b,\mathbf{k}}^{(0)} .
\tag{83}
$$

The entries of $\mathcal{B}^{(1)}$ are easily evaluated noting that $h_{\mathrm{SO}}^{(1)} = \hbar\Omega_R \sigma_x/2$, $\mu^{(1)} = 0$ (see Sec. A.2), and

$$
\begin{aligned}
h_{\mathrm{D}}^{(1)} = & \left[ -\frac{G_{dd}^2 - (4E_R + G_{dd})(4E_R + G_{ss})}{4(2E_R + G_{dd})}\sigma_x - \frac{E_R(3G_{dd} + G_{ss})}{2(2E_R + G_{dd})}(e^{2ix} + e^{-2ix}) \right. \\
& \left. + \frac{G_{dd}(G_{dd} - G_{ss})}{(2E_R + G_{dd})}(e^{4ix}\sigma_+ + e^{-4ix}\sigma_-) \right]\frac{\hbar\Omega_R}{4E_R} ,
\end{aligned}
\tag{84a}
$$

$$
\begin{aligned}
h_{\mathrm{C}}^{(1)} = & \left[ -\frac{(G_{dd} + G_{ss})(4E_R + G_{dd})}{4(2E_R + G_{dd})} - \frac{E_R(G_{dd} - G_{ss})}{2(2E_R + G_{dd})}(e^{2ix} + e^{-2ix})\sigma_x \right. \\
& \left. + \frac{G_{dd}(G_{dd} + G_{ss})}{4(2E_R + G_{dd})}(e^{4ix}\sigma_\uparrow + e^{-4ix}\sigma_\downarrow) \right]\frac{\hbar\Omega_R}{4E_R} .
\end{aligned}
\tag{84b}
$$

The first-order correction to the Bogoliubov frequency is vanishing. To prove this, we first multiply both sides of Eq. (83) by $\mathcal{W}_{b,\mathbf{k}}^{(0)\dagger}$, and use the equality $\mathcal{W}_{b,\mathbf{k}}^{(0)\dagger}\mathcal{B}^{(0)} = \hbar\omega_{b,\mathbf{k}}^{(0)}\mathcal{W}_{b,\mathbf{k}}^{(0)\dagger}\eta$ to eliminate the terms containing $\mathcal{W}_{b,\mathbf{k}}^{(1)}$. Then, we integrate with respect to the spatial coordinates, and use the normalization condition (78a), yielding

$$
\hbar\omega_{b,\mathbf{k}}^{(1)} = \mathcal{B}_{bb,\mathbf{kk}}^{(1)} = 0 .
\tag{85}
$$

In order to calculate the first-order correction to the Bogoliubov amplitudes we first need to evaluate

$$
\int_V d^3r \, \mathcal{W}_{b',\mathbf{k}'}^{(0)\dagger}\eta \mathcal{W}_{b,\mathbf{k}}^{(1)} = \frac{\mathcal{B}_{b'b,\mathbf{k}'\mathbf{k}}^{(1)}}{\hbar[\omega_{b,\mathbf{k}}^{(0)} - \omega_{b',\mathbf{k}'}^{(0)}]} ,
\tag{86a}
$$

$$
\int_V d^3r \, \bar{\mathcal{W}}_{b',\mathbf{k}'}^{(0)\dagger}\eta \mathcal{W}_{b,\mathbf{k}}^{(1)} = \frac{\mathcal{B}_{\bar{b}'b,\overline{\mathbf{k}'}\mathbf{k}}^{(1)}}{\hbar[\omega_{b,\mathbf{k}}^{(0)} + \omega_{b',\mathbf{k}'}^{(0)}]} .
\tag{86b}
$$

These results were obtained multiplying both sides of Eq. (83) by $\mathcal{W}_{b',\mathbf{k}'}^{(0)\dagger}$, integrating over space, and using the orthonormalization conditions (78). Equation (86a) cannot be used when $(b',\mathbf{k}') = (b,\mathbf{k})$; to address this case, we first notice that expanding the normalization condition (31) one finds at first order

$$
\mathrm{Re}\left[ \int_V d^3r \, \mathcal{W}_{b,\mathbf{k}}^{(0)\dagger}\eta \mathcal{W}_{b,\mathbf{k}}^{(1)} \right] = 0 .
\tag{87}
$$

Besides, the imaginary part of this integral can be set equal to zero with an appropriate choice of the phase of $\mathcal{W}_{b,\mathbf{k}}$. Hence, one eventually has $\int_V d^3r\, \mathcal{W}_{b,\mathbf{k}}^{(0)\dagger} \eta \mathcal{W}_{b,\mathbf{k}}^{(1)} = 0$. Using Eq. (82) with $n = 1$ one ends up with

$$\mathcal{W}_{b,\mathbf{k}}^{(1)} = \sum_{b' \neq b} \frac{\mathcal{B}_{b'b,\mathbf{k}\mathbf{k}}^{(1)}}{\hbar[\omega_{b,\mathbf{k}}^{(0)} - \omega_{b',\mathbf{k}}^{(0)}]} \mathcal{W}_{b',\mathbf{k}}^{(0)} - \sum_{b'} \frac{\mathcal{B}_{\overline{b'b},\overline{\mathbf{k}}\mathbf{k}}^{(1)}}{\hbar[\omega_{b,\mathbf{k}}^{(0)} + \omega_{b',-\mathbf{k}}^{(0)}]} \bar{\mathcal{W}}_{b',-\mathbf{k}}^{(0)}. \tag{88}$$

These relations, combined with formulas (101) and (102) in Appendix C, enable one to compute the first-order correction to the amplitude of any Bogoliubov mode.

### B.4 Second-order corrections to the Bogoliubov modes

Equation (80) with $n = 2$ reads

$$\mathcal{B}^{(0)} \mathcal{W}_{b,\mathbf{k}}^{(2)} + \mathcal{B}^{(1)} \mathcal{W}_{b,\mathbf{k}}^{(1)} + \mathcal{B}^{(2)} \mathcal{W}_{b,\mathbf{k}}^{(0)} = \hbar \omega_{b,\mathbf{k}}^{(0)} \eta \mathcal{W}_{b,\mathbf{k}}^{(2)} + \hbar \omega_{b,\mathbf{k}}^{(1)} \eta \mathcal{W}_{b,\mathbf{k}}^{(1)} + \hbar \omega_{b,\mathbf{k}}^{(2)} \eta \mathcal{W}_{b,\mathbf{k}}^{(0)}. \tag{89}$$

Here the operator $\mathcal{B}^{(2)}$ can be evaluated recalling that

$$h_{\mathrm{SO}}^{(2)} = \frac{\hbar^2 k_R k_1^{(2)}}{m} \left( -\nabla_x^2 + i\sigma_z \nabla_x \right), \tag{90}$$

$k_1^{(2)}$ is given by the second term of Eq. (21), $\mu^{(2)}$ by that of Eq. (19), and

$$
\begin{aligned}
h_{\mathrm{D}}^{(2)} = \Bigg[ & -\frac{G_{dd} - G_{ss}}{8} (e^{2ix}\sigma_+ + e^{-2ix}\sigma_-) \\
& + \frac{\left(128 E_R^3 + 48 G_{dd} E_R^2 + 6 G_{dd}^2 E_R + G_{dd}^3\right)(G_{dd} - G_{ss})}{32(2E_R + G_{dd})^2 (8E_R + G_{dd})} (e^{-2ix}\sigma_+ + e^{2ix}\sigma_-) \\
& - \frac{3 G_{dd} E_R (4E_R + G_{dd})(3 G_{dd} + G_{ss})}{8(2E_R + G_{dd})^2 (8E_R + G_{dd})} (e^{4ix} + e^{-4ix}) \\
& + \frac{3 G_{dd}^2 (6E_R + G_{dd})(G_{dd} - G_{ss})}{32(2E_R + G_{dd})^2 (8E_R + G_{dd})} (e^{6ix}\sigma_+ + e^{-6ix}\sigma_-) \Bigg] \left(\frac{\hbar\Omega_R}{4E_R}\right)^2,
\end{aligned} \tag{91a}
$$

$$
\begin{aligned}
h_{\mathrm{C}}^{(2)} = \Bigg[ & -\frac{G_{dd} + G_{ss}}{8} (e^{2ix}\sigma_\uparrow + e^{-2ix}\sigma_\downarrow) \\
& + \frac{\left(128 E_R^3 + 48 G_{dd} E_R^2 + 6 G_{dd}^2 E_R + G_{dd}^3\right)(G_{dd} + G_{ss})}{32(2E_R + G_{dd})^2 (8E_R + G_{dd})} (e^{-2ix}\sigma_\uparrow + e^{2ix}\sigma_\downarrow) \\
& - \frac{3 G_{dd} E_R (4E_R + G_{dd})(G_{dd} - G_{ss})}{8(2E_R + G_{dd})^2 (8E_R + G_{dd})} (e^{4ix} + e^{-4ix})\sigma_x \\
& + \frac{3 G_{dd}^2 (6E_R + G_{dd})(G_{dd} + G_{ss})}{32(2E_R + G_{dd})^2 (8E_R + G_{dd})} (e^{6ix}\sigma_\uparrow + e^{-6ix}\sigma_\downarrow) \Bigg] \left(\frac{\hbar\Omega_R}{4E_R}\right)^2.
\end{aligned} \tag{91b}
$$

The second-order correction to the frequency can be calculated starting from Eq. (89) and performing the same kind of manipulations as those leading to Eq. (85). One eventually finds

$$\hbar \omega_{b,\mathbf{k}}^{(2)} = \mathcal{B}_{bb,\mathbf{k}\mathbf{k}}^{(2)} + \sum_{b' \neq b} \frac{\left|\mathcal{B}_{b'b,\mathbf{k}\mathbf{k}}^{(1)}\right|^2}{\hbar[\omega_{b,\mathbf{k}}^{(0)} - \omega_{b',\mathbf{k}}^{(0)}]} - \sum_{b'} \frac{\left|\mathcal{B}_{\overline{b'b},\overline{\mathbf{k}}\mathbf{k}}^{(1)}\right|^2}{\hbar[\omega_{b,\mathbf{k}}^{(0)} + \omega_{b',-\mathbf{k}}^{(0)}]}, \tag{92}$$

where we made use of Eqs. (85) and (88). This expression can be explicitly evaluated using the formulas in Appendix C. In general, for a given Bogoliubov mode the correction to the frequency (92) contains up to nine nonzero terms.

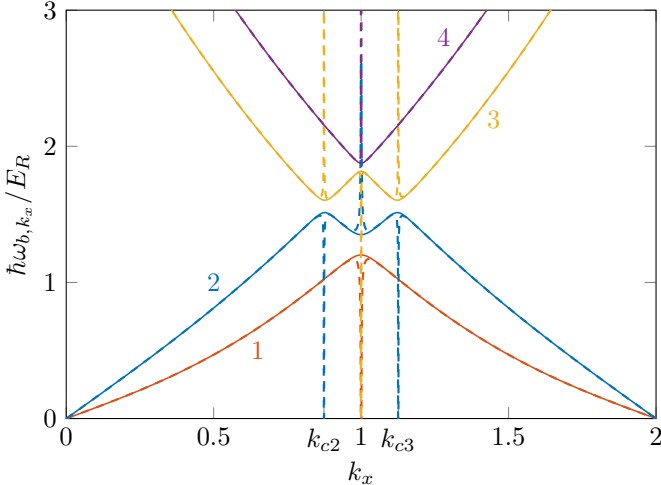

Figure 6: Lowest bands of the Bogoliubov spectrum as functions of $k_x$ at $\hbar\Omega_R = 0.2\,E_R$. The solid curves represent the exact values determined numerically, the dashed ones correspond to the perturbation estimates. Note the unphysical divergence of the latter at $k_x = k_{c2}, 1, k_{c3}$. Close to each band we write the corresponding value of the index $b$. The interaction parameters are the same as Fig. 1.

In Fig. 6 we compare the numerically computed spectrum with the results of our perturbation approach up to second order. We consider excitations propagating along $x$ and show the lowest four bands for a sufficiently small value of the Raman coupling. The agreement is excellent at almost any $k_x$, except in the proximity of the modes where two bands cross at $\Omega_R = 0$ (black dots of Fig. 5), where Eq. (92) predicts an unphysical divergent behavior (see also the discussion at the end of Sec. 4.2). This effect becomes stronger at increasing $\Omega_R$.

For the computation of the two sound velocities we take the two lowest-lying bands ($b = 1, 2$) in the $k \to 0$ limit; recall that at $\Omega_R = 0$ these bands correspond to $\ell = s, d$ and $\bar{K} = 0$ (see Sec. B.1). Each of the nine terms entering the expression (92) of the frequency correction goes like $1/k$ as $k \to 0$. However, after summing all of them the infrared divergence is canceled, and the final result is linear in $k$, $\omega_{\ell,0,\mathbf{k}} \sim c_\ell(\theta_\mathbf{k})k$ with

$$c_\ell(\theta_\mathbf{k}) = c_{\ell,x} \cos^2\theta_\mathbf{k} + c_{\ell,\perp} \sin^2\theta_\mathbf{k} \tag{93}$$

and $\theta_\mathbf{k}$ the angle between $\mathbf{k}$ and the positive $x$ direction. The values of the $c_{\ell,x}$'s and $c_{\ell,\perp}$'s are given by Eqs. (44) and (45) of the main text, respectively. Notice that the scaling (33) of $\mathbf{k}$ requires an analogous transformation of the sound velocity, $c_\ell \to c_\ell/(k_1^2 \cos^2\theta_\mathbf{k} + k_R^2 \sin^2\theta_\mathbf{k})^{1/2}$, so to leave $\omega_{\ell,0,\mathbf{k}}$ unaffected. In writing Eqs. (44) and (45) we inverted this transformation so to obtain an undistorted value of the $c_\ell$'s.

The second-order correction to the Bogoliubov amplitudes is given by Eq. (82) with $n = 2$ and the expansion coefficients

$$
\int_V d^3r\, \mathcal{W}_{b',\mathbf{k}'}^{(0)\dagger} \eta \mathcal{W}_{b,\mathbf{k}}^{(2)} = \frac{\mathcal{B}_{b'b,\mathbf{k}'\mathbf{k}}^{(2)}}{\hbar[\omega_{b,\mathbf{k}}^{(0)} - \omega_{b',\mathbf{k}'}^{(0)}]} + \sum_{(b'',\mathbf{k}'')\neq(b,\mathbf{k})} \frac{\mathcal{B}_{b'b'',\mathbf{k}'\mathbf{k}''}^{(1)}}{\hbar[\omega_{b,\mathbf{k}}^{(0)} - \omega_{b',\mathbf{k}'}^{(0)}]} \frac{\mathcal{B}_{b''b,\mathbf{k}''\mathbf{k}}^{(1)}}{\hbar[\omega_{b,\mathbf{k}}^{(0)} - \omega_{b'',\mathbf{k}''}^{(0)}]}
$$
$$
- \sum_{b'',\mathbf{k}''} \frac{\mathcal{B}_{b'\overline{b''},\mathbf{k}'\overline{\mathbf{k}''}}^{(1)}}{\hbar[\omega_{b,\mathbf{k}}^{(0)} - \omega_{b',\mathbf{k}'}^{(0)}]} \frac{\mathcal{B}_{b''b,\overline{\mathbf{k}''}\mathbf{k}}^{(1)}}{\hbar[\omega_{b,\mathbf{k}}^{(0)} + \omega_{b'',\mathbf{k}''}^{(0)}]}, \tag{94}
$$

$$\int_V d^3r\, \bar{\mathcal{W}}^{(0)\dagger}_{b',\mathbf{k}'} \eta \mathcal{W}^{(2)}_{b,\mathbf{k}} = \frac{\mathcal{B}^{(2)}_{\overline{b'b},\overline{\mathbf{k}'}\mathbf{k}}}{\hbar[\omega^{(0)}_{b,\mathbf{k}} + \omega^{(0)}_{b',\mathbf{k}'}]} + \sum_{(b'',\mathbf{k}'')\neq(b,\mathbf{k})} \frac{\mathcal{B}^{(1)}_{\overline{b'b''},\overline{\mathbf{k}'}\mathbf{k}''}}{\hbar[\omega^{(0)}_{b,\mathbf{k}} + \omega^{(0)}_{b',\mathbf{k}'}]} \frac{\mathcal{B}^{(1)}_{b''b,\mathbf{k}''\mathbf{k}}}{\hbar[\omega^{(0)}_{b,\mathbf{k}} - \omega^{(0)}_{b'',\mathbf{k}''}]}$$
$$- \sum_{b'',\mathbf{k}''} \frac{\mathcal{B}^{(1)}_{\overline{b'b''},\mathbf{k}'\mathbf{k}''}}{\hbar[\omega^{(0)}_{b,\mathbf{k}} + \omega^{(0)}_{b',\mathbf{k}'}]} \frac{\mathcal{B}^{(1)}_{\overline{b''b},\overline{\mathbf{k}''}\mathbf{k}}}{\hbar[\omega^{(0)}_{b,\mathbf{k}} + \omega^{(0)}_{b'',\mathbf{k}''}]}. \tag{95}$$

The derivation of these formulas is analogous to that of Eqs. (86) for the first-order corrections. As in the latter case, when $(b',\mathbf{k}') = (b,\mathbf{k})$ one has to replace Eq. (94) by the relation

$$\int_V d^3r\, \mathcal{W}^{(0)\dagger}_{b,\mathbf{k}} \eta \mathcal{W}^{(2)}_{b,\mathbf{k}} = -\frac{1}{2} \sum_{(b',\mathbf{k}')\neq(b,\mathbf{k})} \frac{\left|\mathcal{B}^{(1)}_{b'b,\mathbf{k}'\mathbf{k}}\right|^2}{\hbar^2[\omega^{(0)}_{b,\mathbf{k}} - \omega^{(0)}_{b',\mathbf{k}'}]^2} + \frac{1}{2} \sum_{b',\mathbf{k}'} \frac{\left|\mathcal{B}^{(1)}_{\overline{b'b},\overline{\mathbf{k}'}\mathbf{k}}\right|^2}{\hbar^2[\omega^{(0)}_{b,\mathbf{k}} + \omega^{(0)}_{b',\mathbf{k}'}]^2}, \tag{96}$$

which follows from the expansion of the normalization condition (31) up to second order [with the phase of $\mathcal{W}_{b,\mathbf{k}}$ again chosen such that the coefficient (96) be real].

In Sec. 4.3 we studied the low-$k$ behavior of the amplitudes of the two gapless Bogoliubov bands up to first order in $\hbar\Omega_R/4E_R$, see Eqs. (46). The general form of these amplitudes at arbitrary Raman coupling is

$$\mathcal{W}_{d,0,\mathbf{k}} = \begin{pmatrix} U_{d,0,\mathbf{k}} \\ V_{d,0,\mathbf{k}} \end{pmatrix} \underset{k\to 0}{\sim} \frac{1}{2i}\sqrt{\frac{2mc_{d,\perp}}{\hbar k N}} \left[ \alpha_d(\theta_\mathbf{k}) \begin{pmatrix} i\Psi_0 \\ -i\Psi_0^* \end{pmatrix} + \beta_d(\theta_\mathbf{k})k_R^{-1} \begin{pmatrix} \nabla_x \Psi_0 \\ \nabla_x \Psi_0^* \end{pmatrix} \right], \tag{97a}$$

$$\mathcal{W}_{s,0,\mathbf{k}} = \begin{pmatrix} U_{s,0,\mathbf{k}} \\ V_{s,0,\mathbf{k}} \end{pmatrix} \underset{k\to 0}{\sim} \frac{1}{2i}\sqrt{\frac{2mc_{s,\perp}}{\hbar k N}} \left[ \alpha_s(\theta_\mathbf{k})k_R^{-1} \begin{pmatrix} \nabla_x \Psi_0 \\ \nabla_x \Psi_0^* \end{pmatrix} + \beta_s(\theta_\mathbf{k}) \begin{pmatrix} i\Psi_0 \\ -i\Psi_0^* \end{pmatrix} \right]. \tag{97b}$$

Here we use dimensional coordinates and momenta. The coefficients $\alpha_\ell$ and $\beta_\ell$ are real and depend on $\theta_\mathbf{k}$, $\Omega_R$, and the interaction parameters. The structures (97) show that, because of the spin-orbit coupling, the density and spin modes do not exhibit a pure superfluid or crystal behavior, rather both characters are present even at low $k$. The only exception is for excitations propagating perpendicular to the $x$ axis ($\theta_\mathbf{k} = \pi/2$), for which $\beta_\ell = 0$ and the hybridization mechanism does not occur. At second order in $\hbar\Omega_R/4E_R$ the coefficients entering Eqs. (97) have the following expressions:

$$\alpha_\ell(\theta_\mathbf{k}) = \alpha_{\ell,x}\cos^2\theta_\mathbf{k} + \alpha_{\ell,\perp}\sin^2\theta_\mathbf{k}, \quad \beta_\ell(\theta_\mathbf{k}) = \beta_{\ell,x}\cos\theta_\mathbf{k}, \tag{98}$$

with $\alpha_{d,\perp} = 1$,

$$\alpha_{d,x} = 1 + \frac{E_R(8E_R^2 + 2E_R G_{ss} + 4E_R G_{dd} + G_{dd}^2)}{4(2E_R + G_{dd})^2(2E_R + G_{ss})}\left(\frac{\hbar\Omega_R}{4E_R}\right)^2, \tag{99a}$$

$$\alpha_{s,x} = 1 + \frac{24E_R^3 + 22E_R^2 G_{dd} + 15E_R G_{dd}^2 + 3G_{dd}^3}{4(2E_R + G_{dd})^3}\left(\frac{\hbar\Omega_R}{4E_R}\right)^2, \tag{99b}$$

$$\alpha_{s,\perp} = 1 + \frac{8E_R^2 + 4E_R G_{dd} + G_{dd}^2}{4(2E_R + G_{dd})^2}\left(\frac{\hbar\Omega_R}{4E_R}\right)^2, \tag{99c}$$

and

$$\sqrt{\frac{G_{dd}}{2E_R}}\beta_{d,x} = -\sqrt{\frac{G_{ss}}{2E_R}}\beta_{s,x}$$
$$= \frac{2E_R G_{dd}\left[2E_R^2 G_{ss} + \left(2E_R^2 + 5E_R G_{ss} + G_{ss}^2\right)G_{dd} + (E_R + G_{ss})G_{dd}^2\right]}{(2E_R + G_{dd})^3(2E_R + G_{ss})(G_{dd} - G_{ss})}\left(\frac{\hbar\Omega_R}{4E_R}\right)^2. \tag{100}$$

The values of the $\alpha$'s and the $\beta$'s as functions of $\Omega_R$ are plotted in Fig. 7, where only excitations propagating parallel or perpendicular to $x$ are considered. The above perturbative estimates are in excellent agreement with the exact numerical values up to significantly large values of the Raman coupling. In particular, in Fig. 7(c) it is shown that the property $\alpha_d(\theta_\mathbf{k} = \pi/2) = 1$ holds at arbitrary $\Omega_R$, even beyond the perturbative regime.

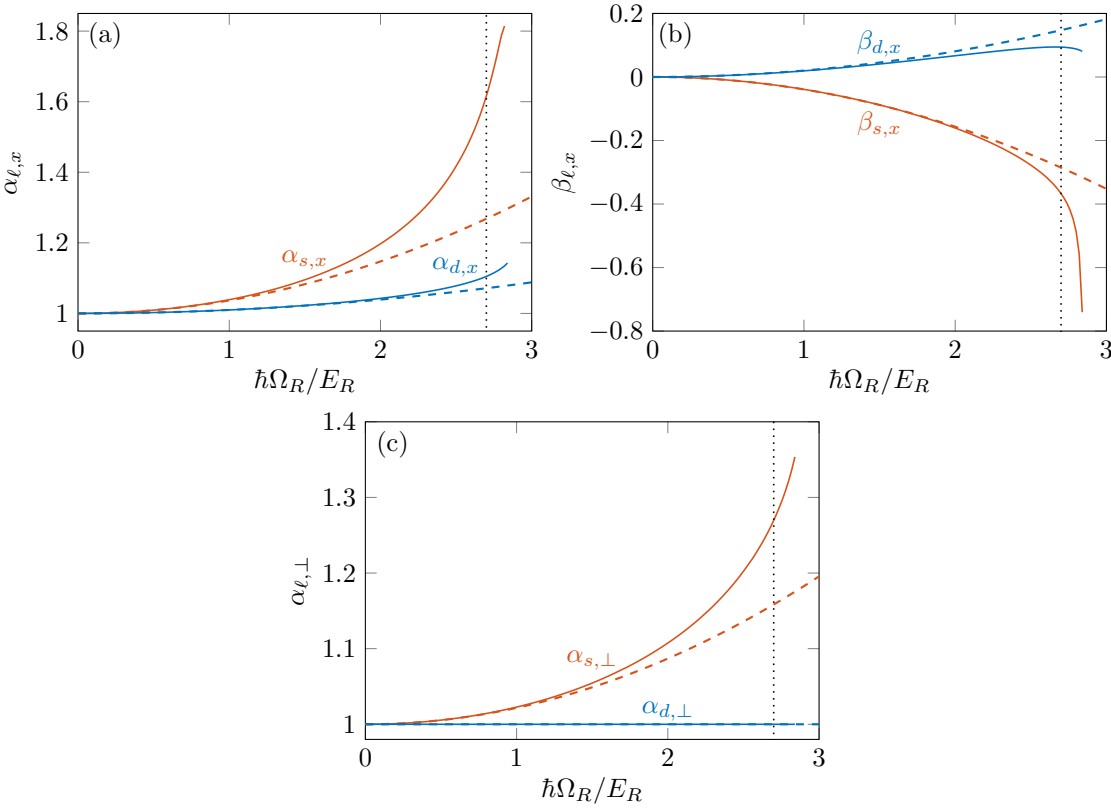

Figure 7: Coefficients of Eqs. (97) as functions of the Raman coupling $\Omega_R$. We show the results for excitations propagating [(a), (b)] parallel and (c) perpendicular to the $x$ axis (in the latter case only the $\alpha$'s are nonvanishing). The solid curves represent the exact numerical values, the dashed ones correspond to the perturbation results obtained for the density (blue) and spin (red) gapless mode. The vertical dotted lines mark the critical Raman coupling $\hbar\Omega_R = 2.70\,E_R$ at which the transition to the plane-wave phase takes place. The interaction parameters are the same as Fig. 1.

## C  Integrals involving the Bogoliubov matrix $\mathcal{B}$

The calculation of the perturbative corrections to the Bogoliubov frequencies and amplitudes (see Appendix B) requires the knowledge of the quantities (81). For $n = 1$ all the $\mathcal{B}^{(1)}_{b'b,\mathbf{kk}}$'s can be expressed in terms of the four integrals

$$
\int_V d^3r\, \mathcal{W}^\dagger_{d,\bar{K}\pm2,\mathbf{k}} \mathcal{B}^{(1)} \mathcal{W}_{d,\bar{K},\mathbf{k}}
$$

$$
= \frac{\hbar\Omega_R}{4(2E_R + G_{dd})} \left[ (E_R - G_{dd})\sqrt{\frac{\Omega_{\bar{K},\mathbf{k}}\Omega_{\bar{K}\pm2,\mathbf{k}}}{\omega_{d,\bar{K},\mathbf{k}}\omega_{d,\bar{K}\pm2,\mathbf{k}}}} + E_R\sqrt{\frac{\omega_{d,\bar{K},\mathbf{k}}\omega_{d,\bar{K}\pm2,\mathbf{k}}}{\Omega_{\bar{K},\mathbf{k}}\Omega_{\bar{K}\pm2,\mathbf{k}}}} \right], \tag{101a}
$$

$$
\int_V d^3r\, \mathcal{W}^\dagger_{s,\bar{K}\pm2,\mathbf{k}} \mathcal{B}^{(1)} \mathcal{W}_{s,\bar{K},\mathbf{k}}
$$

$$
= -\frac{\hbar\Omega_R}{4(2E_R + G_{dd})} \left[ (E_R + G_{dd} + G_{ss})\sqrt{\frac{\Omega_{\bar{K},\mathbf{k}}\Omega_{\bar{K}\pm2,\mathbf{k}}}{\omega_{s,\bar{K},\mathbf{k}}\omega_{s,\bar{K}\pm2,\mathbf{k}}}} + (E_R + G_{dd})\sqrt{\frac{\omega_{s,\bar{K},\mathbf{k}}\omega_{s,\bar{K}\pm2,\mathbf{k}}}{\Omega_{\bar{K},\mathbf{k}}\Omega_{\bar{K}\pm2,\mathbf{k}}}} \right],
$$

$$
\tag{101b}
$$

$$
\int_V d^3r\, \mathcal{W}^\dagger_{s,\bar{K}\pm2,\mathbf{k}} \mathcal{B}^{(1)} \mathcal{W}_{d,\bar{K},\mathbf{k}}
$$

$$
= \mp\frac{\hbar\Omega_R}{16E_R} \left[ (2E_R - G_{dd})\sqrt{\frac{\Omega_{\bar{K},\mathbf{k}}\omega_{s,\bar{K}\pm2,\mathbf{k}}}{\omega_{d,\bar{K},\mathbf{k}}\Omega_{\bar{K}\pm2,\mathbf{k}}}} + (2E_R + G_{ss})\sqrt{\frac{\omega_{d,\bar{K},\mathbf{k}}\Omega_{\bar{K}\pm2,\mathbf{k}}}{\Omega_{\bar{K},\mathbf{k}}\omega_{s,\bar{K}\pm2,\mathbf{k}}}} \right], \tag{101c}
$$

$$
\int_V d^3r\, \mathcal{W}^\dagger_{d,\bar{K}\pm2,\mathbf{k}} \mathcal{B}^{(1)} \mathcal{W}_{s,\bar{K},\mathbf{k}}
$$

$$
= \pm\frac{\hbar\Omega_R}{16E_R} \left[ (2E_R + G_{ss})\sqrt{\frac{\Omega_{\bar{K},\mathbf{k}}\omega_{d,\bar{K}\pm2,\mathbf{k}}}{\omega_{s,\bar{K},\mathbf{k}}\Omega_{\bar{K}\pm2,\mathbf{k}}}} + (2E_R - G_{dd})\sqrt{\frac{\omega_{s,\bar{K},\mathbf{k}}\Omega_{\bar{K}\pm2,\mathbf{k}}}{\Omega_{\bar{K},\mathbf{k}}\omega_{d,\bar{K}\pm2,\mathbf{k}}}} \right]. \tag{101d}
$$

All the other integrals of this kind are vanishing. Similarly, in order to compute the $\mathcal{B}^{(1)}_{b'b,-\mathbf{kk}}$'s one needs to know

$$
\int_V d^3r\, \bar{\mathcal{W}}^\dagger_{d,-(\bar{K}\pm2),-\mathbf{k}} \mathcal{B}^{(1)} \mathcal{W}_{d,\bar{K},\mathbf{k}}
$$

$$
= \frac{\hbar\Omega_R}{4(2E_R + G_{dd})} \left[ (E_R - G_{dd})\sqrt{\frac{\Omega_{\bar{K},\mathbf{k}}\Omega_{-(\bar{K}\pm2),-\mathbf{k}}}{\omega_{d,\bar{K},\mathbf{k}}\omega_{d,-(\bar{K}\pm2),-\mathbf{k}}}} - E_R\sqrt{\frac{\omega_{d,\bar{K},\mathbf{k}}\omega_{d,-(\bar{K}\pm2),-\mathbf{k}}}{\Omega_{\bar{K},\mathbf{k}}\Omega_{-(\bar{K}\pm2),-\mathbf{k}}}} \right], \tag{102a}
$$

$$
\tag{102b}
$$

$$
\int_V d^3r\, \bar{\mathcal{W}}^\dagger_{s,-(\bar{K}\pm2),-\mathbf{k}} \mathcal{B}^{(1)} \mathcal{W}_{s,\bar{K},\mathbf{k}}
$$

$$
= -\frac{\hbar\Omega_R}{4(2E_R + G_{dd})} \left[ (E_R + G_{dd} + G_{ss})\sqrt{\frac{\Omega_{\bar{K},\mathbf{k}}\Omega_{-(\bar{K}\pm2),-\mathbf{k}}}{\omega_{s,\bar{K},\mathbf{k}}\omega_{s,-(\bar{K}\pm2),-\mathbf{k}}}} - (E_R + G_{dd})\sqrt{\frac{\omega_{s,\bar{K},\mathbf{k}}\omega_{s,-(\bar{K}\pm2),-\mathbf{k}}}{\Omega_{\bar{K},\mathbf{k}}\Omega_{-(\bar{K}\pm2),-\mathbf{k}}}} \right],
$$

$$\int_V d^3r\, \bar{\mathcal{W}}^\dagger_{s,-(\bar{K}\pm 2),-\mathbf{k}} \mathcal{B}^{(1)} \mathcal{W}_{d,\bar{K},\mathbf{k}}$$

$$= \pm \frac{\hbar\Omega_R}{16 E_R} \left[ (2E_R - G_{dd}) \sqrt{\frac{\Omega_{\bar{K},\mathbf{k}}\,\omega_{s,-(\bar{K}\pm 2),-\mathbf{k}}}{\omega_{d,\bar{K},\mathbf{k}}\,\Omega_{-(\bar{K}\pm 2),-\mathbf{k}}}} - (2E_R + G_{ss}) \sqrt{\frac{\omega_{d,\bar{K},\mathbf{k}}\,\Omega_{-(\bar{K}\pm 2),-\mathbf{k}}}{\Omega_{\bar{K},\mathbf{k}}\,\omega_{s,-(\bar{K}\pm 2),-\mathbf{k}}}} \right], \tag{103a}$$

$$\int_V d^3r\, \bar{\mathcal{W}}^\dagger_{d,-(\bar{K}\pm 2),-\mathbf{k}} \mathcal{B}^{(1)} \mathcal{W}_{s,\bar{K},\mathbf{k}}$$

$$= \mp \frac{\hbar\Omega_R}{16 E_R} \left[ (2E_R + G_{ss}) \sqrt{\frac{\Omega_{\bar{K},\mathbf{k}}\,\omega_{d,-(\bar{K}\pm 2),-\mathbf{k}}}{\omega_{s,\bar{K},\mathbf{k}}\,\Omega_{-(\bar{K}\pm 2),-\mathbf{k}}}} - (2E_R - G_{dd}) \sqrt{\frac{\omega_{s,\bar{K},\mathbf{k}}\,\Omega_{-(\bar{K}\pm 2),-\mathbf{k}}}{\Omega_{\bar{K},\mathbf{k}}\,\omega_{d,-(\bar{K}\pm 2),-\mathbf{k}}}} \right]. \tag{103b}$$

Analogous expressions can be found for the $\mathcal{B}^{(1)}_{b'\bar{b},-\mathbf{k}\mathbf{k}}$'s and the $\mathcal{B}^{(1)}_{b'\bar{b},-\overline{\mathbf{k}\mathbf{k}}}$'s. For $n = 2$ we give here only the results for the coefficients $\mathcal{B}^{(2)}_{bb,\mathbf{k}\mathbf{k}}$ entering the second-order correction to the frequency (92):

$$\int_V d^3r\, \mathcal{W}^\dagger_{d,\bar{K},\mathbf{k}} \mathcal{B}^{(2)} \mathcal{W}^{(0)}_{d,\bar{K},\mathbf{k}}$$

$$= \frac{k_1^{(2)}}{k_R} \frac{\omega_{d,\bar{K},\mathbf{k}}^2 + \Omega_{\bar{K},\mathbf{k}}^2}{\omega_{d,\bar{K},\mathbf{k}}\Omega_{\bar{K},\mathbf{k}}} E_R (k_x + \bar{K})^2 + \frac{G_{dd}(G_{dd}+G_{ss})}{8\hbar\omega_{d,\bar{K},\mathbf{k}}} \left( \frac{\hbar\Omega_R}{4E_R} \right)^2 \tag{104a}$$

$$+ \frac{32 E_R^3 + 12 G_{dd} E_R^2 - G_{dd}^3 + (2E_R + G_{dd})^2 G_{ss}}{[4(2E_R + G_{dd})]^2} \left( \frac{\hbar\Omega_R}{4E_R} \right)^2 \frac{\omega_{d,\bar{K},\mathbf{k}}^2 + \Omega_{\bar{K},\mathbf{k}}^2}{\omega_{d,\bar{K},\mathbf{k}}\Omega_{\bar{K},\mathbf{k}}},$$

$$\int_V d^3r\, \mathcal{W}^\dagger_{s,\bar{K},\mathbf{k}} \mathcal{B}^{(2)} \mathcal{W}_{s,\bar{K},\mathbf{k}}$$

$$= \frac{k_1^{(2)}}{k_R} \frac{\omega_{s,\bar{K},\mathbf{k}}^2 + \Omega_{\bar{K},\mathbf{k}}^2}{\omega_{s,\bar{K},\mathbf{k}}\Omega_{\bar{K},\mathbf{k}}} E_R (k_x + \bar{K})^2 + \frac{G_{ss}(G_{dd}+G_{ss})}{8\hbar\omega_{s,\bar{K},\mathbf{k}}} \left( \frac{\hbar\Omega_R}{4E_R} \right)^2 \tag{104b}$$

$$+ \frac{32 E_R^3 + 20 G_{dd} E_R^2 + 8 G_{dd}^2 E_R + G_{dd}^3 - (2E_R + G_{dd})^2 G_{ss}}{[4(2E_R + G_{dd})]^2} \left( \frac{\hbar\Omega_R}{4E_R} \right)^2 \frac{\omega_{s,\bar{K},\mathbf{k}}^2 + \Omega_{\bar{K},\mathbf{k}}^2}{\omega_{s,\bar{K},\mathbf{k}}\Omega_{\bar{K},\mathbf{k}}}.$$

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
