# Peer review of "Supersolid phase of a spin-orbit-coupled Bose-Einstein condensate: a perturbation approach"

_SciPost Physics, doi:SciPost Phys. 11, 092 (2021)_

## Round 1 · Referee Report · Ian Spielman (Referee 1) · 2021-8-16

Report

Supersolidity was first conceived in the the context of condensed matter physics. In that setting distinguishing a supersolid from either a solid or a superfluid relies on bulk responses. For example a system with non-zero sheer modulus (solid behavior) that supports a supercurrent would be a very strong supersolid candidate. Indeed initial reports of supersolidity (now thought to be an out of equilibrium state) in 4He used a torsional oscillator to measure just this.

By contrast, cold atom experiments probing supersolid systems focus on the common abstract definition of a supersolid as "a system with coexisting long-range diagonal (lattice structure) and off-diagonal order (coherence)".

Martone and Stringari address this question in the context of spin-orbit Bose-Einstein condensates (SOBECs) where these questions are more nuanced. Here the presence of crystalline order coexisting with long-range coherence in the so-called "stripe" phase is established. Still the very tunability of cold-atom experiments makes the unambiguous association of the striped phase with a supersolid more challenging. The crux of the matter is that when the Raman coupling creating the spin orbit coupling (SOC) is removed the stripe phase is still present (but without its tell-tale stripes!) as a uniformly mixed binary superfluid. Furthermore when stripes are present their period is very nearly given by parameters governed by single particle physics (an atomic superfluid in an optical lattice should not be accepted as a supersolid even though it possesses long range diagonal and off-diagonal order -- more on this below).

This work addressed the question of supersolidty in a third way by focusing on the two low-energy branches of collective excitation spectrum, showing that one branch corresponds to a lattice phonon (vibrations of the stripes) and the other to a superfluid phonon (corresponding to excitations of the overall phase). Only one branch would be present in the optical lattice case where the density structure is externally imposed and does not have an associated phonon mode.

Lastly the authors use sum rules to derive a new way to extract the superfluid fraction as a ratio of longitudinal and transverse velocities. The sum rules expressions give bulk response functions, and while there does not appear to be an analog to the sheer modulus commented on above, but they still find a useful relation linking compressibility with speed of sound and superfluid density.

This work is well written and should be published with little revision.

Comments and discussion

In many of these cases I will be contrasting with the binary spin mixture case to attempt to draw out essential differences between the $\Omega_R = 0$ miscible/immiscible phase transition, which continuously connects to the stripe/planewave transition in the $g_{ss} / \Omega_R$ plane.

  • On page 6: I found the comment on the metastable stripe phase up to a spinodal point interesting. I have two questions about this. Is this saying that a quench from the ST to the PW phase will be stable until crossing the spinodal point? At which points it will exhibit a dynamical instability for the growth of spin structure? In this formalism is such a spinodal point present in the binary mixture case? I would think not.

  • In the context of Eq. (9) I think it is unclear in context that $k_1$ is amongst the parameters that must be adjusted to find the ground state. This is clarified lower on the page as well as on the next page, but it seems an important point.

  • On page 8: The anzatz of Eq. (15) has problems when $g_{ss} < 0$ (or at least would require a different $\Psi_0$). This leads to the question (which I believe is established elsewhere): does the stripe phase ever exist for $g_{ss} < 0$ (when the binary mixture is immiscible).

  • On page 14: As the manuscript progresses we are faced with expressions like 44a and 44b that are difficult to make physical sense out of. Is there something that can be said about the various groups of coefficients that carry meaning?

  • In Eq. (55) does the $q \rightarrow -q$ indicate a second term created by replacing $q$ with $-q$ in the first term? I have never seen this sort of notation before.

  • In general, the results presented use parameters from a past theoretical study, rather than $^{87}{\rm Rb}$, where the majority of the relevant experiments have take place. To ease comparison with experiment it would be desirable to have some key figures using experimental parameters.

  • As a closing question, if we consider the $g_{ss}/ \Omega_R$ plane (including negative $\Omega_R$) how would the authors label the singular group of states at $\Omega_R = 0$? The evidence presented is strong that we should label all states as supersolids.

  • validity: -
  • significance: -
  • originality: -
  • clarity: -
  • formatting: -
  • grammar: -

Author:  Giovanni Italo Martone  on 2021-10-01  [id 1793]

(in reply to Report 1 by Ian Spielman on 2021-08-16)

We are indebted to Prof. Spielman for his insightful remarks and positive assessment of our manuscript. We provide below our response to his points.

$*$ Concerning the problem of metastable stripes, we answer affirmatively to the first question: if one ramps up the Raman coupling across the transition from the ST to the PW phase, the stripe phase remains dynamically stable even past the transition, up to the spinodal point, where the velocity of spin waves vanishes, revealing the occurrence of a dynamic instability associated with the divergence of magnetic susceptibility. In the miscible-immiscible transition at $\Omega_R = 0$ the metastability window shrinks to a single value $g_{ss} = 0$, such that the transition and spinodal points coincide. We have expanded the discussion of the behaviour of the sound velocities in Sec. 4.3 to cover these points.

$*$ We have added a short sentence below Eq. (9) to make readers immediately aware that $k_1$ is a parameter to be adjusted to find the ground state.

$*$ Regarding the existence of the stripe phase in the $g_{ss} < 0$ regime, it was indeed ruled out by the variational analysis of Ref. [53]. We now mention this result in the section devoted to the phase diagram, right below Eq. (8).

$*$ Being the results of a perturbation approach, expressions like (44a) and (44b) of the previous submission are quite cumbersome and their coefficients do not have a direct and simple physical meaning. The only thing one can learn from them is that the spin-orbit coupling makes the dependence of the sound velocities on the interaction parameters considerably more involved than in standard binary mixtures. We have added this comment to the text below the formulae.

$*$ We have rewritten Eq. (55) of the previous submission without the $q \to - q$ notation for better clarity.

$*$ In this manuscript we indeed use different parameters from those of ${}^{87}$Rb, in particular we take a much larger value of the ratio $g_{ss}/g_{dd}$. The reason is that, if one uses the parameters of ${}^{87}$Rb, the stripe phase exist in a very narrow range of values of the Raman coupling, $0 \leq \hbar\Omega_R \lesssim 0.19 \, E_R$. At such low values of $\Omega_R$ the supersolidity effects predicted in our work are exceedingly small, and physical quantities are practically undistinguishable from their $\Omega_R = 0$ values. For this reason we preferred not to make new figures, but we added a discussion below Eq. (21) about the issues of the ${}^{87}$Rb parameters, which was missing in the previous submission.

$*$ The last question on how to label the singular group of states at $\Omega_R = 0$ is very interesting. We have expanded Sec. 3.1 to address this point. In short, the $\Omega_R = 0$ states are not supersolid because they have no density modulations, and thus no crystalline order. The $\Omega_R = 0$ states should rather be classified as easy-plane ferromagnets: they have polarization lying in the $(x,y)$ plane, whose direction is fixed by a mechanism of spontaneous breaking of spin-rotation invariance about the $z$ axis. The spin Goldstone mode then describes a rotation of the polarization direction. This is in stark contrast with the situation at finite $\Omega_R$, where density fringes appear giving rise to a genuine crystal Goldstone mode associated with the breaking of translation invariance, see Eq. (48b).

---

## Round 1 · Referee Report · Anonymous (Referee 2) · 2021-9-2

Report

Dear Editor,

In the present manuscript authors aim to investigate Goldstone modes of a Bose-Einstein condensate (BEC) in the presence of spin-orbit coupling. Actually, over the last ten years the subject has been deeply investigated both experimentally and using various mean-filed approximations. For this reason, the physics for these kind of systems is already quite well known. The main improvement I found in the present study is the application of a perturbative approach (up to the second order in the Raman coupling) in order to probe Eq.(12). As a result, they are able to show the stabilisation of stripe phases for certain parameters’s values which, on the other hand, do not seem particularly helpful for experiments.

Stripe phase is indeed an interesting topic in this moment being observable in many experimental systems. “Dipolar atoms” (dysprosium or erbium) provide virtuous examples. However I must say that the “feature” of supersolidity is not always easy to spot by means of a mean-field methods. Despite authors gave a good introduction on the subject, they should also mention that, differently from three-dimensional cases, a dipolar system in two-dimension does not show any stripe phase sustaining supersolidity in its ground state. Evidence of this has been recently given using numerical methodologies, such as quantum Monte Carlo, that go beyond a mean-filed approach. It worth to include those findings in the introduction.

I am not completely sure that authors can talk about supersolidity for the stripe phase they obtained with their approximations. Referring to figure 4 superfluidity along the x-direction is mainly one for ℏΩR/ER ≲2. In this region, if their calculations underestimate superfluity, I will expect that a rigours treatment of quantum fluctuations may show a melting of stripes in favour of an homogeneous superfluid phase. On the contrary, if they are overestimating superfluidity a real phase coherence between stripes should disappear at T=0. Therefore I suspect that the present perturbative analysis does not furnish a clear evidence of a supersolid phase. Undoubtedly, a method that would take into account quantum fluctuation would settle the matter definitively. Since they are only dealing with mean-filed approximations, I recommend to include in section 4.5 the evaluation of the “Leggett’s criterion” for the stripe regime at least. A comparison between the Leggett’s methods and the perturbation approach will help to understand if supersolidity might last using these set of parameters.

To conclude I deem that the present work may only result a slight improvement to the understanding of spin-orbit-coupled BECs. In any case, from a technical perspective, it could be useful for the community of theoretical physicists that master mean-field approaches. I suggest the publication only after having included my considerations into the main text.
  • validity: ok
  • significance: ok
  • originality: low
  • clarity: high
  • formatting: good
  • grammar: good

Author:  Giovanni Italo Martone  on 2021-10-01  [id 1794]

(in reply to Report 2 on 2021-09-02)

We thank the referee for carefully reviewing our work and providing us valuable suggestions to improve it. Below we summarise our response to their criticism.

$*$ In the introduction of their report the referee affirms that the parameter values employed in our work "do not seem particularly helpful for experiments". We have slightly reformulated the discussion below Eq. (21) to provide stronger motivations for our choice of the interaction parameters, which were lacking in the previous version of our manuscript. It is true that our parameters are very different from those of ${}^{87}$Rb (see also the response to referee 1) because of the larger ratio $g_{ss} / g_{dd}$ that we consider. However, such conditions can be reached in experiments by reducing the spatial overlap between the two spin or pseudo-spin components. This idea has already been implemented in the experimental work [10] by Ketterle's group, employing pseudo-spin states generated by a superlattice and resulting in the detection of stripes. Alternatively one can use atomic species such as ${}^{39}$K, where several Feshbach resonances are available. We thus believe that future studies will be able to test our new predictions.

$*$ We agree that the subject of stripes in two-dimensional dipolar gases deserves to be mentioned in the introduction. We have consequently provided additional references in the manuscript.

$*$ In the third paragraph of their report the referee raises the point of the role of quantum fluctuations in our system, which might preclude supersolidity either favouring an homogeneous phase or suppressing the phase coherence between the stripes. We first point out that, unlike the case of dipolar gases, quantum fluctuation effects are typically negligible in dilute three-dimensional spin-orbit-coupled BECs. This can be inferred by looking at the value of the quantum depletion, which in previous works [50,51] was found to be of the order of a few percent; we checked that this is still the case for the values of the parameters employed in our article.

$*$ The referee proposes us to compare our results for the superfluid density with the predictions of the Leggett criterion. This is an excellent suggestion and we added the related discussion to the manuscript. Leggett's criterion was originally derived assuming a class of Ansatz many-body wave functions in which all the particles in the superfluid have the same phase. The mean-field wave function belongs to this class, hence one would expect Leggett's upper bound for the superfluid density to be saturated in mean-field treatments. However, our system is peculiar because it lacks Galilean invariance, which is another requirement for the derivation of Leggett's criterion. As a consequence, the superfluid fraction obtained from this criterion is generally larger than the actual value. This effect is visible in the stripe phase and more strikingly in the uniform ones, where Leggett's criterion does not predict any depletion of the superfluid fraction, whereas the approach of Ref. [62], which takes explicitly into account the effects caused by the breaking of Galilean invariance, yields a much smaller value (see Fig. 4).

$*$ In the conclusion of their report the referee states that our work "may only result a slight improvement to the understanding of spin-orbit-coupled BECs". In this respect we stress that, in addition to providing analytical formulas for physical quantities in the stripe phase, our work clarifies important conceptual points never addressed in the previous literature, including the connection between the spin branch of the Bogoliubov spectrum and the crystal Goldstone mode, and the relation between the superfluid density and the sound velocity. For all these reasons we do believe that the scientific quality of our manuscript meets the editorial standards required by SciPost Physics.

---

## Round 2 · Author Response

We thank both referees for their careful review of our manuscript, which has been revised to address their remarks. In addition we have made several minor improvements, including a short discussion of the hybridisation of the crystal and superfluid Goldstone modes in the stripe phase (accounted for by second-order corrections in $\hbar \Omega_R / 4 E_R$ to the mode wave functions). We hope that the improved version of our work meets the criteria of interest and importance required for publication in SciPost Physics.

---

## Round 2 · List of Changes

Changes in response to remarks by Prof. Spielman: - Below Eq. (8) we have briefly commented on the phase diagram for $g_{ss} < 0$; - Below Eq. (9) we now state that $k_1$ is a parameter to be adjusted to find the ground state; - In Sec. 3.1 we have enlarged the discussion on the characterisation of the $\Omega_R = 0$ states; to this purpose we have introduced the arbitrary phase $\chi_0$ in the formulas of this section (taken equal to $0$ in the calculations of the Appendices); - In the paragraph below Eqs. (19)-(21) we have commented on supersolid effects in current ${}^{87}$Rb experiments; - Below Eqs. (44) and (45) we have commented on the involved dependence of the sound velocity on the interaction parameters, and we have discussed the occurrence of a dynamic instability at the spinodal point; - On page 17 we have improved the discussion on the spin Goldstone mode at $\Omega_R = 0$ (also recalled in the conclusion); - We have rewritten Eq. (55) of the previous version eliminating the $q \to - q$ notation.

Changes in response to remarks by Referee 2: - In the introduction we have added a comment on the stripe phase of two-dimensional dipolar gases; - At the beginning of Sec. 2.2 we have added a comment on the quantum depletion in our system; - In the paragraph below Eqs. (19)-(21) we have improved the discussion on the experimental relevance of our results; - On page 21 we now mention the availability of Monte Carlo calculations in our system (Ref. [54]); - Again on page 21 we have added a paragraph comparing our results for the superfluid density with those of the Leggett criterion, and added the corresponding curves in Fig. 4 (a short comment has also been added in the conclusion).

Additional changes: - In the introduction we have quoted further experimental [17,18,27] and theoretical [32,33] results; - Below Eq. (6) we now explicitly state that the stability conditions $g > 0$ and $g_{dd} > 0$ are assumed; - In Eq. (26) we now take $\lambda$ complex to eliminate the arbitrary phase $\phi$ introduced in Eqs. (40) of the previous submission; - In Sec. 4.5 we have restored some missing mass factors; - On page 20 we now state that the superfluid density in our system is an anisotropic tensor and that Eq. (60) holds in the symmetric intraspecies coupling case; - In App. B.4 we have added a discussion and a new figure on the hybridisation of the crystal and superfluid modes in the stripe phase; correspondingly, the paragraph below Eqs. (46) in the main text has been amended; - A few typos in the previous submission have been fixed; - References [19,37] of the previous submission have been updated.

---

## Editorial Decision

published